# Joint Learning of Label and Environment Causal Independence for Graph Out-of-Distribution Generalization

**Shurui Gui,   Meng Liu,   Xiner Li,   Youzhi Luo,   Shuiwang Ji**
Department of Computer Science & Engineering
Texas A&M University
College Station, TX 77843
`{shurui.gui,mengliu,lxe,yzluo,sji}@tamu.edu`

## Abstract

We tackle the problem of graph out-of-distribution (OOD) generalization. Existing graph OOD algorithms either rely on restricted assumptions or fail to exploit environment information in training data. In this work, we propose to simultaneously incorporate label and environment causal independence (LECI) to fully make use of label and environment information, thereby addressing the challenges faced by prior methods on identifying causal and invariant subgraphs. We further develop an adversarial training strategy to jointly optimize these two properties for causal subgraph discovery with theoretical guarantees. Extensive experiments and analysis show that LECI significantly outperforms prior methods on both synthetic and real-world datasets, establishing LECI as a practical and effective solution for graph OOD generalization. Our code is available at https://github.com/divelab/LECI.

## 1  Introduction

Graph learning methods have been increasingly used for many diverse tasks, such as drug discovery [1], social network analysis [2], and physics simulation [3]. One of the major challenges in graph learning is out-of-distribution (OOD) generalization, where the training and test data are typically from different distributions, resulting in significant performance degradation when deploying models to new environments. Despite several recent efforts [4–7] have been made to tackle the OOD problem on graph tasks, their performances are often not satisfactory due to the complicated distribution shifts on graph topology and the violations of their assumptions or premises in reality. In addition, while environment-based methods have shown success in traditional OOD fields [8, 9], they cannot perform favorably on graphs [10]. Several existing graph environment-centered methods [11–13] aim to *infer environment labels* for graph tasks, however, most methods *utilize the pre-collected environment labels* in a manner similar to traditional environment-based techniques [9, 14, 15], rather than exploiting them in a unique and specific way tailored for graphs. Therefore, the potential exploitation of environment information in graph OOD tasks remains limited (Sec. 2.2). Although annotating or extracting environment labels requires additional cost, it has been proved that generalization without extra information is theoretically impossible [16] (Appx. B.2). This is analogous to the impossibility of inferring or approximating causal effects without counterfactual/intervened distributions [17, 18].

In this paper, we propose a novel learning strategy to incorporate label and environment causal independence (LECI) for tackling the graph OOD problem. Enforcing such independence properties can help exploit environment information and alleviate the challenging issue of graph topology shifts. Specifically, our contributions are summarized below. (1) We identify the current causal subgraph discovery challenges and propose to solve them by introducing label and environment causal independence. We further present a practical solution, an adversarial learning strategy, to

37th Conference on Neural Information Processing Systems (NeurIPS 2023).

jointly optimize such two causal independence properties with causal subgraph discovery guarantees. (2) LECI is positioned as the first graph-specific pre-collected environment exploitation learning strategy. This learning strategy is applicable to any subgraph discovery networks and environment inference methods. (3) According to our extensive experiments, LECI outperforms baselines on both the structure/feature shift sanity checks and real-world scenario comparisons. With additional visualization, hyperparameter sensitivity, training dynamics, and ablation studies, LECI is empirically proven to be a practically effective method. (4) Contrary to prevalent beliefs and results, we showcase that pre-collected environment information, far from being useless, can be a potent tool in graph tasks, which is evidenced more powerful than previous assumptions.

## 2   Background

### 2.1   Graph OOD generalization

We represent an attributed graph as $G = (X, A) \in \mathcal{G}$, where $\mathcal{G}$ is the graph space. $X \in \mathbb{R}^{n \times d}$ and $A \in \mathbb{R}^{n \times n}$ denote its node feature matrix and adjacency matrix respectively, where $n$ is the number of nodes and $d$ is the feature dimension. In the graph-level OOD generalization setting, each graph $G$ has an associated label $Y \in \mathcal{Y}$, where $\mathcal{Y}$ denotes the label space. Notably, training and test data are typically drawn from different distributions, *i.e.*, $P^{tr}(G, Y) \neq P^{te}(G, Y)$. Unlike the image space, where distribution shifts occur only on feature vectors, distribution shifts in the graph space can happen on more complicated variables such as graph topology [19, 20]. Therefore, many existing graph OOD methods [4–7, 11] aim to identify the most important topological information, called causal subgraphs, so that they can be used to make predictions that are robust to distribution shifts.

### 2.2   Environment-based OOD algorithms

To address distribution shifts, following invariant causal predictor (ICP) [21] and invariant risk minimization (IRM) [9], many environment-based invariant learning algorithms [14, 22–24], also referred as parts of domain generalization (DG) [25–27], classify data into several groups called environments, denoted as $E \in \mathcal{E} = \{e_1, e_2, \ldots, e_{|\mathcal{E}|}\}$. Intuitively, data in the same environment group share similar uncritical information, such as the background of images and the size of graphs. It is assumed that the shift between the training and test distribution should be reflected among environments. Therefore, to address generalization problems in the test environments, these methods incorporate the environment shift information from training environments into neural networks, thereby driving the networks to be invariant to the shift between the training and test distribution.

Several works further consider the inaccessibility of environment information in OOD settings. For example, environment inference methods [11, 12, 28] propose to infer environment labels to make invariant predictions. While in graph tasks, several methods [4–6, 13] attempt to generalize without the use of environments. However, these algorithms usually rely on relatively strict assumptions that can be compromised in many real-world scenarios and may be more difficult to satisfy than the access of environment information, which will be further compared and justified in Appx C.2. As demonstrated in the Graph Out-of-Distribution (GOOD) benchmark [10] and DrugOOD [29], the environment information for graph datasets is commonly accessible. Thus, in this paper, we assume the availability of environment labels and focus on invariant predictions by exploiting the given environment information in graph-specific OOD settings. Extensive discussions of related works and comparisons to previous graph OOD methods are available in Appx B.

**Comparisons to other environment-based graph OOD methods.** It's crucial to distinguish between the two stages of environmental-based methods: environmental inference and environmental exploitation. The first phase, environmental inference, involves predicting environmental labels, while the second, environmental exploitation, focuses on using pre-acquired environmental labels. Typically, an environmental inference method employs an environmental exploitation method to evaluate its effectiveness. However, an environmental exploitation method does not require an environmental inference method. Recent developments in graph-level environmental inference methods [11, 12] and node-level environmental inference methods [13] introduce graph-specific environmental inference techniques. Nevertheless, their corresponding environmental exploitation strategies are not tailored to graphs. In contrast, our method, LECI, bypasses the environmental inference phase and instead introduces a graph-specific environmental exploitation algorithm, supported by justifications in Appx B.3. More environment-related discussions and motivations are available in Appx. B.2.

# 3 Method

OOD generalization is a longstanding problem because different distribution shifts exist in many different applications of machine learning. As pointed out by Kaur et al. [30], understanding the data generation process is critical for solving OOD problems. Hence, we first formalize the target problems through three topology distribution shift assumptions from a causal and data generation perspective. We further identify the challenges of discovering causal subgraphs for addressing topology distribution shifts. Building on our analysis, we propose a technical solution and a practical implementation to jointly optimize label and environment causal independence in order to learn causal subgraphs effectively.

## 3.1 Causal perspective of graph OOD generalization

A major difference between traditional OOD and graph OOD tasks is the topological structure distribution shift in graph OOD tasks. To analyze graph structure shifts, we commonly assume that only a part of a graph determines its target $Y$; *e.g.*, only the functional motif of a molecule determines its corresponding property. Therefore, in graph distribution shift studies, we assume that each graph $G \in \mathcal{G}$ is composed of two subgraphs; namely, a causal subgraph $G_C \in \mathcal{G}$ and a spurious subgraph $G_S \in \mathcal{G}$ as shown in the structure causal models (SCMs) [17, 18] of Fig. 1. For notational convenience, we denote graph union and subtraction operations as $+$ and $-$, respectively; *e.g.*

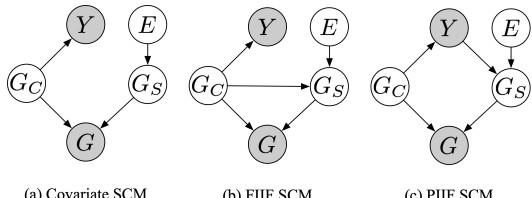

(a) Covariate SCM      (b) FIIF SCM      (c) PIIF SCM

Figure 1: Illustrations of **three distribution shift assumptions.** In these structural causal models (SCMs), each node denotes a variable, and each directional edge represents causation. Grey nodes are observable variables that can be directly accessed, but it does not mean that we are "conditioned on" these variables.

$G = G_C + G_S$ and $G_C = G - G_S$. It follows from the above discussions that $G_C$ is the only factor determining the target variable $Y \in \mathcal{Y}$, while $G_S$ is controlled by an exogenous environment $E$, such that graphs in the same environment share similar spurious subgraphs.

As illustrated in Fig. 1 (a), the covariate shift represents the most common distribution shift. Similar to image problems in which the same object is present with different backgrounds, graphs that share the same functional motif along with diverse graph backbones belong to the covariate shift. Under this setting, $G_S$ is only controlled by $E$. Models trained in this setting suffer from distribution shifts because the general empirical risk minimization (ERM) training objective $P(Y|G)$ is conditioned on the collider $G$, which builds a spurious correlation between $G_S$ and $Y$ through $Y \leftarrow G_C \rightarrow G \leftarrow G_S$, where $\rightarrow$ denotes a causal correlation. The fully informative invariant features (FIIF) and the partially informative invariant features (PIIF) assumptions, shown in Fig. 1 (b) and (c), are two other common assumptions proposed by invariant learning studies [6, 9, 31]. In the graph FIIF assumption, $G_S$ is controlled by both $G_C$ and $E$, which constructs an extra spurious correlation $Y \leftarrow G_C \rightarrow G_S$. In contrast, the graph PIIF assumption introduces an anti-causal correlation between $G_S$ and $Y$. Our work focuses on the covariate shift generalization to develop our approach and then extends the solution to both FIIF and PIIF assumptions.

Essentially, we are addressing the distribution shifts between $P^{tr}(G, Y)$ and $P^{te}(G, Y)$. Our core assumption, based on the Independence Causal Mechanism (ICM), is that there exists a component $G_C$ within $G$ such that $P^{tr}(Y|G_C) = P^{te}(Y|G_C)$. The shifts in distribution are exclusively attributed to interventions on $G_S = G - G_C$ that are encapsulated as an environment variable $E$, aligned with the invariant learning literature. The OOD dilemma arises when $P^{tr}(E) \neq P^{te}(E)$, leading to shifts in both $P^{tr}(G_S)$ and $P^{te}(G_S)$, and consequentially, $P^{tr}(G) \neq P^{te}(G)$. It's crucial to note that we assume both $P^{tr}(G_C)$ and $P^{te}(G_C)$ retain the same support. Explicitly, our target is to resolve OOD scenarios where the supports of $P^{tr}(E)$ and $P^{te}(E)$ differ.

## 3.2 Subgraph discovery challenges

A common strategy to make neural networks generalizable to structure distribution shifts in graph OOD tasks is to identify causal subgraphs for invariant predictions [4–6, 19]. However, correctly selecting subgraphs is challenging due to the following two precision challenges. (1) The selected causal subgraph may contain spurious structures, *i.e.*, $\exists G_p \subseteq G_S, G_p \subseteq \hat{G}_C$, where $G_a \subseteq G_b$

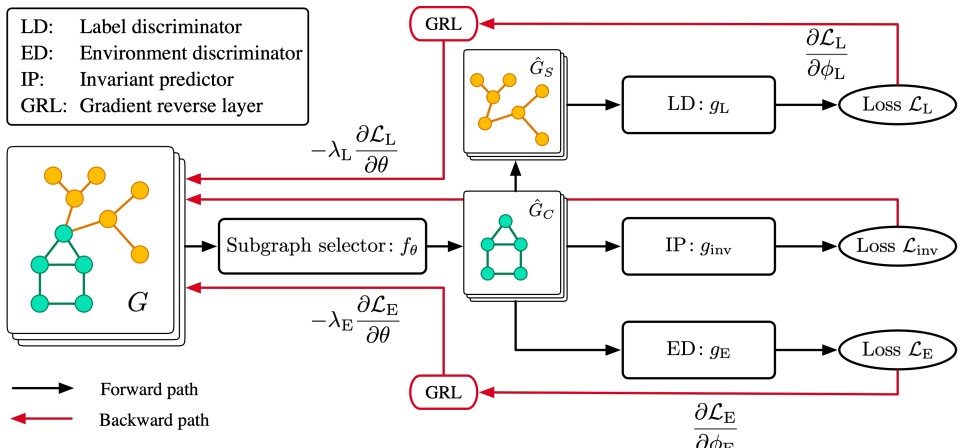

Figure 2: An illustration of the invariant prediction process by selecting the causal subgraph $\hat{G}_C$ using an interpretable subgraph discovery network in LECI. This network and training details are described in Sec. 3.4. We multiply the reversed gradients with two hyperparameters $\lambda_L$ and $\lambda_E$ to control the adversarial training. For clear notations, we define the three losses $\mathcal{L}_{inv}$, $\mathcal{L}_L$, and $\mathcal{L}_E$ in Eq. 1, 7, and 6, respectively.

denotes $G_a$ is a subgraph of $G_b$, and $\hat{G}_C$ represents the selected causal subgraph. (2) The model may not select the whole $G_C$ into $\hat{G}_C$ and leave a part of $G_C$ in $\hat{G}_S = G - \hat{G}_C$. Formally, $\exists G_p \subseteq G_C, G_p \subseteq \hat{G}_S$. The reason for the occurrence of these precision challenges will be further discussed in Appx. C.3.

### 3.3 Two causal independence properties

To address the precision issues, it is necessary to distinguish between causal subgraphs and spurious subgraphs, both theoretically and in practice. To achieve this, under the covariate assumption, we introduce two causal independence properties for causal and spurious subgraphs, respectively; those are $E \perp\!\!\!\perp G_C$ and $Y \perp\!\!\!\perp G_S$.

The first property $E \perp\!\!\!\perp G_C$ is a crucial consideration to alleviate the first precision problem. As illustrated in Fig. 1 (a), since $G$ acts as a collider that blocks the correlation between $E$ and $G_C$, the environment factor $E$ should be independent of the causal subgraph $G_C$. Conversely, due to the direct causation between $E$ and $G_S$, the spurious subgraph $G_S$ is highly correlated with $E$. This correlation difference with $E$ indicates that enforcing independence between $E$ and the selected causal subgraph $\hat{G}_C$ can prevent parts of spurious subgraphs in $\hat{G}_C$ be included, thereby solving the first precision problem. Note that this independence also holds in FIIF and PIIF settings, because $G_S$ acts as another collider and blocks the left correlation paths between $G_C$ and $E$, *i.e.*, $G_C \to G_S \leftarrow E$ and $G_C \to Y \to G_S \leftarrow E$.

We propose to enforce the second independence property $Y \perp\!\!\!\perp G_S$ in order to address the second precision problem. Under the covariate assumption, the correlation between $Y$ and $G_S$ is blocked by $G$, leading to the independence between $Y$ and $G_S$. On the other side, the causal subgraph $G_C$ is intrinsically correlated with $Y$. Similarly, this correlation difference with $Y$ motivates us to enforce such independence, thus filtering parts of causal subgraphs out of the selected spurious subgraphs $\hat{G}_S$. This property $Y \perp\!\!\!\perp G_S$, however, does not hold under the FIIF and PIIF settings, because of the $Y \leftarrow G_C \to G_S$ and the $Y \to G_S$ correlations. Therefore, we introduce a relaxed version of the independence property: for any $G_p$ that $G_S \subseteq G_p$, $I(G_S; Y) \leq I(G_p; Y)$ (See Appx. C.1.1).

### 3.4 Adversarial implementation

The above two causal independence properties are intuitively helpful in discovering causal subgraphs. In this section, we present our practical solution to learn the label and environment causal independence (LECI). The overall architecture of LECI is illustrated in Fig. 2. To be specific, we use an interpretable subgraph discovery network as the basic architecture, which consists of a subgraph selector $f_\theta : \mathcal{G} \mapsto \mathcal{G}$ and an invariant predictor $g_{inv} : \mathcal{G} \mapsto \mathcal{Y}$. Here, $f_\theta$ and $g_{inv}$ model the distributions $P_\theta(G_C|G)$ and $P_{\phi_{inv}}(Y|\hat{G}_C)$, where $\phi_{inv}$ is the model parameters. Specifically, following Miao et al.

[5], our subgraph selector module selects subgraphs by sampling edges from the original graphs. However, the sampling process is not differentiable and blocks the gradient back-propagation. To overcome this problem, we use Gumbel-Sigmoid [32] to bypass the non-differentiable sampling process. The basic training loss for this subgraph discovery network can be written as

$$\mathcal{L}_{\text{inv}} = -\mathbb{E}\left[\log P_{\phi_{\text{inv}}}(Y|\hat{G}_C)\right]. \tag{1}$$

It is worth noting that $\hat{G}_C = f_\theta(G)$ and $\hat{G}_S = G - \hat{G}_C$ are complimentary. More details of the subgraph selector can be found in Appx. D.1

To incorporate the causal independence properties described in Sec. 3.3, we first enforce the condition $\hat{G}_C \perp\!\!\!\perp E$. Since $\hat{G}_C \perp\!\!\!\perp E$ is equivalent to $I(E; \hat{G}_C) = 0$, and $I(E; \hat{G}_C) \geq 0$, the objective $I(E; \hat{G}_C)$ reaches its minimum when and only when $\hat{G}_C \perp\!\!\!\perp E$. This leads us to define minimizing $I(E; \hat{G}_C)$ as our training criterion.

**Definition 3.1.** The environment independence training criterion is

$$\theta_{\text{E}}^* = \arg\min_\theta I(E; \hat{G}_C). \tag{2}$$

For a fixed $\theta$, $I(E; \hat{G}_C)$ is essentially a constant. However, the mutual information $I(E; \hat{G}_C) = \mathbb{E}\left[\log \frac{P(E|\hat{G}_C)}{P(E)}\right]$ cannot be calculated directly, since $P(E|\hat{G}_C)$ is unknown. Following Proposition 1 in GAN [33], we introduce an optimal discriminator $g_{\text{E}} : \mathcal{G} \mapsto \mathcal{E}$ with parameters $\phi_E$ to approximate $P(E|\hat{G}_C)$ as $P_{\phi_{\text{E}}}(E|\hat{G}_C)$, minimizing the negative log-likelihood $-\mathbb{E}\left[\log P_{\phi_{\text{E}}}(E|\hat{G}_C)\right]$. We then have the following two propositions:

**Proposition 3.2.** *For $\theta$ fixed, the optimal discriminator $\phi_E$ is*

$$\phi_E^* = \arg\min_{\phi_E} -\mathbb{E}\left[\log P_{\phi_{\text{E}}}(E|\hat{G}_C)\right]. \tag{3}$$

**Proposition 3.3.** *Denoting KL-divergence as $KL[\cdot\|\cdot]$, for $\theta$ fixed, the optimal discriminator $\phi_E$ is $\phi_E^*$, s.t.*

$$KL\left[P(E|\hat{G}_C)\|P_{\phi_E^*}(E|\hat{G}_C)\right] = 0. \tag{4}$$

Proposition 3.2 can be proved straightforwardly by applying the cross-entropy training criterion, while the proof of Proposition 3.3 is provided in Appx C. With these propositions, the mutual information can be computed with the help of the optimal discriminator $\phi_E^*$. According to Proposition 3.3, we have:

$$\begin{aligned} I(E; \hat{G}_C) &= \mathbb{E}\left[\log P_{\phi_{\text{E}}^*}(E|\hat{G}_C)\right] + H(E) + \text{KL}\left[P(E|\hat{G}_C)\|P_{\phi_{\text{E}}^*}(E|\hat{G}_C)\right] \\ &= \mathbb{E}\left[\log P_{\phi_{\text{E}}^*}(E|\hat{G}_C)\right] + H(E) + 0. \end{aligned} \tag{5}$$

Thus, by disregarding the constant $H(E)$, the training criterion becomes:

$$\theta_{\text{E}}^* = \arg\min_\theta I(E; \hat{G}_C) = \arg\min_\theta \mathbb{E}\left[\log P_{\phi_{\text{E}}^*}(E|\hat{G}_C)\right] = \arg\min_\theta \left\{\max_{\phi_E} \mathbb{E}\left[\log P_{\phi_{\text{E}}}(E|\hat{G}_C)\right]\right\}. \tag{6}$$

We can reformulate this training criterion in terms of a negative log-likelihood form ($\mathcal{L}_{\text{E}}$) to define our environment adversarial training criterion (EA):

$$\Theta_{\text{E}}^* = \left\{\theta_{\text{E}}^* : \theta_{\text{E}}^* \in \arg\max_\theta \{\min_{\phi_E} \mathcal{L}_{\text{E}}\}\right\} = \left\{\theta_{\text{E}}^* : \theta_{\text{E}}^* \in \arg\max_\theta \left\{\min_{\phi_E} -\mathbb{E}\left[\log P_{\phi_{\text{E}}}(E|\hat{G}_C)\right]\right\}\right\}. \tag{7}$$

To enforce $\hat{G}_S \perp\!\!\!\perp Y$, we introduce a label discriminator $g_{\text{L}} : \mathcal{G} \mapsto \mathcal{Y}$ to model $P(Y|\hat{G}_S)$ with parameters $\phi_{\text{L}}$ as $P_{\phi_{\text{L}}}(Y|\hat{G}_S)$. This leads to a symmetric label adversarial training criterion (LA):

$$\Theta_{\text{L}}^* = \left\{\theta_{\text{L}}^* : \theta_{\text{L}}^* \in \arg\max_\theta \{\min_{\phi_L} \mathcal{L}_{\text{L}}\}\right\} = \left\{\theta_{\text{L}}^* : \theta_{\text{L}}^* \in \arg\max_\theta \left\{\min_{\phi_L} -\mathbb{E}\left[\log P_{\phi_{\text{L}}}(Y|\hat{G}_S)\right]\right\}\right\}. \tag{8}$$

It's important to note that $\Theta_{\text{E}}^*$ and $\Theta_{\text{L}}^*$ are not unique optimal parameters. Instead, they are two optimal parameter sets, including parameters that satisfy the training criteria. We will discuss $\Theta_{\text{E}}^*$ and $\Theta_{\text{L}}^*$, along with a relaxed version of LA, in further detail in the theoretical analysis section 3.5.

Intuitively, as illustrated in Fig. 2, the models $g_E$ and $g_L$, acting as an environment discriminator and a label discriminator, are optimized to minimize the negative log-likelihoods. In contrast, the subgraph selector $f_\theta$ tries to adversarially maximize the losses by reversing the gradients during back-propagation. Through this adversarial training process, we aim to select a causal subgraph $\hat{G}_C$ that is independent of the environment variables $E$, while simultaneously eliminating causal information from $\hat{G}_S$ such that the selected causal subgraph $\hat{G}_C$ can retain as much causal information as possible for the accurate inference of the target variable $Y$.

## 3.5 Theoretical results

Under the framework of the three data generation assumptions outlined in Section 3.1, we provide theoretical guarantees to address the challenges associated with subgraph discovery. Detailed proofs for the following lemmas and theorems can be found in Appx. C. We initiate our analysis with a lemma for $\hat{G}_C \perp\!\!\!\perp E$:

**Lemma 3.4.** *Given a subgraph $G_p$ of input graph $G$ and SCMs (Fig. 1), it follows that $G_p \subseteq G_C$ if and only if $G_p \perp\!\!\!\perp E$.*

This proof hinges on the data generation assumption that any substructures external to $G_C$ (i.e., $G_S$) maintain associations with $E$. Consequently, any $\hat{G}_C$ that fulfills $\hat{G}_C \perp\!\!\!\perp E$ will be a subgraph of $G_C$, denoted as $\hat{G}_C \subseteq G_C$. We then formulate the EA training lemma as follows:

**Lemma 3.5.** *Given SCMs (Fig. 1), the predicted $\hat{G}_C = f_\theta(G)$, and the EA training criterion (Eq. 6), $\hat{G}_C \subseteq G_C$ if and only if $\theta \in \Theta_E^*$.*

This lemma paves the way for an immediate conclusion: given that $\hat{G}_C$ is not unique under the EA training criterion, the optimal $\Theta_E^*$ is likewise non-unique, rather $\Theta_E^*$ represents a set of parameters. Any $\theta \in \Theta_E^*$ adheres to the environment independence property. Similarly, under the covariate SCM assumption, when we enforce $\hat{G}_S \perp\!\!\!\perp Y$, $\Theta_L^*$ represents an optimal parameter set satisfying the label independence property.

**Lemma 3.6.** *Given the covariate SCM assumption and the LA criterion (Eq. 7), $\hat{G}_S \subseteq G_S$ if and only if $\theta \in \Theta_L^*$.*

Our first subgraph discovery guarantee is a direct corollary of Lemma 3.5 and Lemma 3.6.

**Theorem 3.7.** *Under the covariate SCM assumption and both EA and LA training criteria (Eq. 6 and 7), it follows that $\hat{G}_C = G_C$ if and only if $\theta \in \Theta_E^* \cap \Theta_L^*$.*

However, the aforementioned theorem does not ensure causal subgraph discovery under FIIF and PIIF SCMs. Hence, we need to consider the relaxed property of $\hat{G}_S \perp\!\!\!\perp Y$ discussed in Sec. 3.3 to formulate a more robust subgraph discovery guarantee.

**Theorem 3.8.** *Given the covariate/FIIF/PIIF SCM assumptions and both EA and LA training criteria (Eq. 6 and 7), it follows that $G_C = \hat{G}_C$ if and only if, under the optimal EA training, the LA criterion is optimized, i.e.,$\theta \in \arg\max_{\theta \in \Theta_E^*} \{\min_{\phi_L} \mathcal{L}_L\}$.*

The proof intuition hinges on the assumption that $G_S \subseteq \hat{G}_S$ under the premise of $\hat{G}_C \perp\!\!\!\perp E$. Hence, taking into account the relaxed property, the mutual information $I(\hat{G}_S; Y)$ can be minimized when $\hat{G}_S = G_S$, *i.e.*, $\hat{G}_C = G_C$. This, in turn, suggests that the LA optimization should satisfy $\Theta_L^* \subseteq \Theta_E^*$, denoted as $\Theta_L^* = \arg\max_{\theta \subseteq \Theta_E^*} \{\min_{\phi_L} \mathcal{L}_L\}$. As a result, the EA training criterion takes precedence over the LA criterion during training, which is empirically reflected by the relative weights of the hyperparameters in practical experiments.

## 3.6 Pure feature shift consideration

Even though removing the spurious subgraphs can make the prediction more invariant, the proposed method primarily focuses on the graph structure perspective. However, certain types of spurious information may only be present in the node features, referred to as pure feature shifts. Thus, we additionally apply a technique to address the distribution shifts on node features $X$. In particular, we transform the original node features into environment-free node features by removing the environment

information through adversarially training with a small feature environment discriminator. Ideally, after this feature filtering, there are no pure feature shifts left; thus, the remaining shifts can be eliminated by the causal subgraph selection process. The specific details of this pre-transformation step can be found in the Appx. D.2.

### 3.7 Discussion of computational complexity and assumption comparisons

The time complexity of LECI is $O(md + nd^2)$ where $n$, $m$, and $d$ denote the number of nodes, edges, and feature dimensions, respectively. To be more specific, the message-passing neural networks have time complexity $O(md + nd^2)$. Our environment exploitation regularizations have time complexity $O(1)$ without any extra cost. Therefore, the overall time complexity of our LECI is $O(md + nd^2 + 1) = O(md + nd^2)$. OOD generalization performance cannot be universal and is highly correlated to the generalizability of the method's assumptions. Therefore, we provide theory and assumption comparisons with previous works [4–6] in Appx. C.2.

## 4 Experiments

In this section, we conduct extensive experiments to evaluate our proposed LECI. Specifically, we aim to answer the following 5 research questions through our experiments. **RQ1:** Does the proposed method address the previous unsolved structure shift and feature shift problems? **RQ2:** Does the proposed method perform well in complex real-world settings? **RQ3:** Is the proposed method robust across various hyperparameter settings? **RQ4:** Is the training process of the proposed method stable enough under complex OOD conditions? **RQ5:** Are all components in the proposed method important? The comprehensive empirical results and detailed analysis demonstrate that LECI is an effective and practical solution to address graph OOD problems in various settings.

### 4.1 Baselines

We compare our LECI with the empirical risk minimization (ERM), a graph pooling baseline ASAP [34], 4 traditional OOD baselines, and, 4 recent graph-specific OOD baselines. The traditional OOD baselines include IRM [9], VREx [14], DANN [8], and Coral [35], and the 4 graph-specific OOD algorithms are DIR [4], GSAT [5], CIGA [6], and GIL [11]. It is worth noting that the implementation of CIGA we use is CIGAv2 and we provide an environment exploitation comparison with GIL's environment exploitation phase (IGA [15]) on the same setting subgraph discovery network in Appx B.3. Detailed baseline selection justification can be found in Appx. E.3.

### 4.2 Sanity check on synthetic datasets

In this section, we aim to answer RQ1 by comparing our LECI against several baselines, on both structure shift and feature shift datasets. Following the GOOD benchmark [10], we consider the synthetic dataset GOOD-Motif for a structure shift sanity check and the semi-synthetic dataset GOOD-CMNIST for a feature shift sanity check. Dataset details are available in Appx. E.2.

Specifically, each graph in GOOD-Motif is composed of a base subgraph and a motif subgraph. Notably, only the motif part, selected from 3 different shapes, determines its corresponding 3-class classification label. This dataset has two splits, a base split and a size split. For the base split, environment labels control the shapes of base subgraphs, and we target generalizing to unseen base subgraphs in the test set. In terms of the size split, base subgraphs' size scales vary across different environments. In such a split, we aim to generalize from small to large graphs.

As shown in Tab. 1, our method performs consistently better against all baselines. According to the GOOD benchmark, our method is the only algorithm that performs close to the oracle result (92.09%) on the base split. To further investigate the OOD performance of our method under the three SCM assumptions, we create another synthetic dataset, namely Motif for covariate, FIIF, and PIIF (CFP-Motif). To be specific, there are two major differences compared to GOOD-Motif. First, instead of using paths as base subgraphs in the test environment, we produce Dorogovtsev-Mendes graphs [36] as base subgraphs, which can further evaluate the applicability of generalization results. Second, CFP-Motif extends GOOD-Motif with FIIF and PIIF shifts. In FIIF and PIIF splits (Fig. 1), $G_C$ and $Y$ have a probability of 0.9 to determine the size of $G_S$, leading to spurious correlations w.r.t. size. Among the three shifts, PIIF is the hardest one for LECI due to the stronger correlations between $G_S$ and $Y$ than FIIF, which may cause the relaxed version of independence property (Sec. 3.3) less

Table 1: **Results on structure and feature shift datasets.** The reported results are the classification accuracies on test sets with standard deviations in parentheses. All reported results are obtained through an automatic hyperparameter selection process with 3 runs. The best and second-best results are highlighted in **bold** and underline respectively.

| | GOOD-Motif | | GOOD-CMNIST | | CFP-Motif | |
| | basis | size | color | covariate | FIIF | PIIF |
|---|---|---|---|---|---|---|
| ERM | 60.93(11.11) | 56.63(7.12) | 26.64(2.37) | 57.56(9.59) | 37.22(3.70) | 62.45(9.21) |
| IRM | 64.94(4.85) | 54.52(3.27) | 29.63(2.06) | 58.11(5.14) | 44.33(1.52) | 68.34(10.40) |
| VREx | 61.59(6.58) | 55.85(9.42) | 27.13(2.90) | 48.78(7.81) | 34.78(1.34) | 63.33(6.55) |
| Coral | 61.95(10.36) | 55.80(4.05) | 29.21(6.87) | 57.11(8.35) | 42.67(7.09) | 60.33(8.85) |
| DANN | 50.62(4.71) | 46.61(3.78) | 27.86(5.02) | 49.45(8.05) | 43.22(6.64) | 62.56(10.39) |
| ASAP | 45.00(11.66) | 42.23(4.20) | 23.53(0.67) | 60.00(2.36) | 43.34(7.41) | 35.78(0.88) |
| DIR | 34.39(2.02) | 43.11(2.78) | 22.53(2.56) | 44.67(0.00) | 42.00(6.77) | 47.22(8.79) |
| GSAT | 62.27(8.79) | 50.03(5.71) | 35.02(2.78) | 68.22(7.23) | 51.56(6.59) | 61.22(8.80) |
| CIGA | 37.81(2.42) | 51.87 (5.15) | 25.06(3.07) | 56.78(2.99) | 39.11(7.70) | 45.67(7.52) |
| LECI | **84.56(2.22)** | **71.43(1.96)** | **51.80(2.70)** | **83.20(5.89)** | **77.73(3.85)** | **69.40(7.54)** |

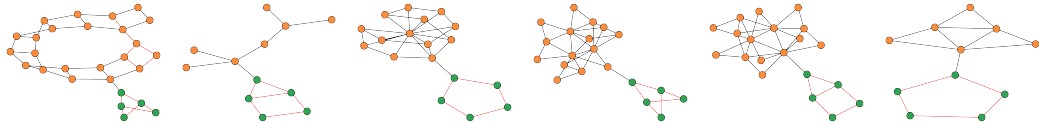

Figure 3: **Interpretability visualization of LECI.** The figure displays the results with the three motifs (crane, house, cycle). The left three graphs are selected from the training set with 3 base graphs, namely, ladder, tree, and wheel, respectively. The right three graphs are from the OOD test set with Dorogovtsev-Mendes graphs as their basis. For clarity, the motifs and base graphs are colored as green and orange nodes, respectively. The selected causal graphs $\hat{G}_C$ are denoted by red edges.

prominent, *i.e.*, larger $I(G_S; Y)$ may lead to smaller gradients from any $G_p$ to $G_S$. Comparing ERM and subgraph discovery OOD methods on GOOD-Motif size and CFP-Motif FIIF/PIIF splits, we observe that although subgraph discovery methods are possible to address size shifts, they are more sensitive than general GNNs.

The feature shift sanity check is performed on GOOD-CMNIST, in which each graph consists of a colored hand-written digit transformed from MNIST via superpixel technique [37]. The environment labels control the digit colors, and the challenge is to generalize to unknown colors in the test set. As reported in Tab. 1, LECI outperforms all baselines by a large margin, indicating the significant improvement of our method on the feature shift problem.

Due to the difficulty of OOD training process, many OOD methods do not achieve their theoretical OOD generalization limits. To further investigate the ability of different learning strategies, we deliberately leak the OOD test set results and apply OOD test set hyperparameter selection to compare the theoretical potentials of different OOD principles in Appx. F.1.

#### 4.2.1 Interpretability visualizations

As illustrated in Fig. 3, LECI can select the motifs accurately, which indicates that LECI eliminates most spurious subgraphs to make predictions. This is the key reason behind LECI's ability to generalize to graphs with different unknown base subgraphs, making it the first method that achieves such invariant predictions on GOOD-Motif. More visualization results are available in the Appx. F.2.

### 4.3 Practical comparisons

In this section, we aim to answer RQ2, RQ3, and RQ4 by conducting experiments on real-world scenarios.

#### 4.3.1 Comparisons on real-world datasets

We cover a diverse set of real-world datasets. For molecular property prediction tasks, we use the scaffold and size splits of GOOD-HIV [10] and the assay split of DrugOOD LBAP-core-ic50 [29] to evaluate our method's performance on different shifts. Additionally, we compare OOD methods on

Table 2: **Real-world scenario performance.** We compare the performance of 10 methods on real-world datasets. The results are reported in terms of accuracy for the first two datasets, and ROC-AUC for the latter two. For each dataset split, ID val and OOD val denote the OOD test set results using the in-distribution and out-of-distribution validation set in each run, repectively.

| | GOOD-SST2 | | GOOD-Twitter | | GOOD-HIV-scaffold | | GOOD-HIV-size | | DrugOOD-assay | |
|---|---|---|---|---|---|---|---|---|---|---|
| | ID val | OOD val | ID val | OOD val | ID val | OOD val | ID val | OOD val | ID val | OOD val |
| ERM | 78.37(2.64) | 80.41 (0.69) | 54.93(0.96) | 57.04(1.70) | 69.61(1.32) | 70.37(1.19) | 61.66(2.45) | 57.31(1.06) | 70.03(0.16) | 72.18(0.18) |
| IRM | 79.73(1.45) | 80.17(1.52) | 55.27(1.19) | 57.72(1.03) | 73.35(2.30) | 70.89(0.29) | 58.52(0.86) | 60.86(2.78) | 71.56(0.32) | 72.69(0.29) |
| VREx | 79.31(1.40) | 80.33(1.09) | 56.46(0.93) | 56.37(0.76) | 71.73(3.51) | 71.18(0.69) | 58.39(1.54) | 60.10(2.09) | 70.22(0.86) | 72.32(0.58) |
| Coral | 78.24(3.26) | 80.97(1.07) | 56.57(0.42) | 56.14(1.76) | 71.19(2.82) | 71.12(2.92) | 60.81(4.76) | 62.07(1.05) | 70.18(0.76) | 72.07(0.56) |
| DANN | 78.74(0.82) | 80.36 (0.61) | 55.52(1.27) | 55.71(1.23) | 69.88(3.66) | 72.25(1.59) | 61.37(0.53) | 60.04(2.11) | 69.83(0.95) | 72.23(0.26) |
| ASAP | 78.51(2.26) | 80.44(0.59) | 56.10(2.65) | 56.37(1.30) | 69.97(2.91) | 68.44(0.49) | 61.08(2.66) | 61.54(2.53) | 68.02(1.22) | 71.73(0.39) |
| DIR | 77.65(0.71) | 81.50(0.55) | 55.32(1.85) | 56.81(0.91) | 65.84(1.71) | 68.59(3.70) | 59.69(1.59) | 60.85(0.52) | 67.29(0.73) | 69.70(0.65) |
| GSAT | 79.25(1.09) | 80.46 (0.38) | 55.09(0.66) | 56.07(0.53) | 71.55(3.58) | 71.39(1.41) | 60.92(1.00) | 60.61(1.19) | 71.01(0.54) | 72.26(0.45) |
| CIGA | 80.37(1.46) | 81.20(0.75) | 57.51(1.36) | 57.19(1.15) | 66.25(2.89) | 71.47(1.29) | 58.24(3.78) | 62.56(1.76) | 67.68(1.14) | 70.54(0.59) |
| LECI | **82.93(0.22)** | **83.44(0.27)** | **59.35(1.44)** | **59.64(0.15)** | **74.04(0.65)** | **74.43(1.69)** | **64.83(2.59)** | **65.44(1.78)** | **72.67(0.46)** | **73.45(0.17)** |

natural language processing datasets, including GOOD-SST2 and Twitter [38]. The Twitter dataset is split similarly to GOOD-SST2, thus it will be denoted as GOOD-Twitter in this paper. According to Tab. 2, LECI achieves the best results over all baselines on real-world datasets. Notably, LECI achieves consistently effective performance regardless of whether the validation set is from the in-distribution (ID) or out-of-distribution (OOD) domain. This indicates the stability of our training process, which will be further discussed in Sec. 4.3.3. Besides, we provide a fairness justification in Appx. E.1.

### 4.3.2 Hyperparameter sensitivity study

In this study, we examine the effect of the hyperparameters $\lambda_E$ and $\lambda_L$ on the performance of our proposed method on the GOOD-Motif and GOOD-Twitter datasets. We vary these hyperparameters around their selected values and observe the corresponding results. As shown in Fig. 4, LECI demonstrates robustness across different hyperparameter settings and consistently outperforms ERM. Notably, since OOD generalizations are harsher than ID tasks, although these results are slightly less stable than ID results, they can be considered as robust in the OOD field compared to previous OOD methods [4, 6]. In these experiments, while we apply stronger independence constraints in synthetic datasets, we use weaker constraints in real-world datasets. This is because the two discriminators are harder to train on real-world datasets, so we slow down the adversarial training of the subgraph selector to maintain the performance of the discriminators.

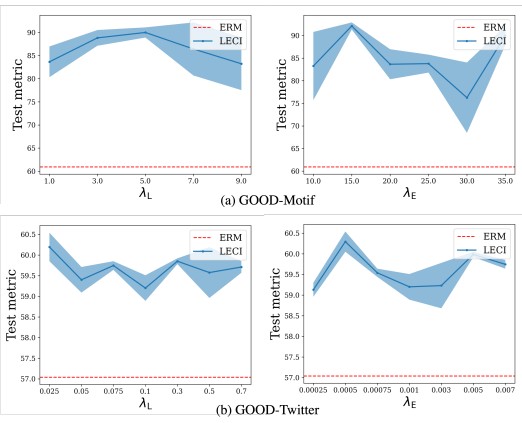

Figure 4: **Sensitivity analysis of $\lambda_L$ and $\lambda_E$.** For each sensitivity study, we fix other hyperparameters with the values selected from the previous experiments.

More information on the hyperparameters can be found in the Appx. E.5.

### 4.3.3 Training stability and dynamics study

The training stability is an important consideration to measure whether a method is practical to implement in real-world scenarios or not. To study LECI's training stability, we plot the OOD test accuracy curves for our LECI and the baselines on GOOD-Motif and GOOD-SST2 during the training processes. As illustrated in Fig. 5, many baselines achieve their highest performances at relatively early epochs but eventually degenerate to worse results after overfitting to the spurious information. On GOOD-Motif, several general OOD baselines converge to accuracy around 70% higher than many other baselines, but these results are sub-optimal since 70% accuracy indicates that these methods nearly misclassify one whole class given the dataset only has 3 classes. In contrast, LECI's OOD test accuracy begins to climb up rapidly once the discriminators are well-trained and the independence constraints take effect. Then, LECI consistently converges to the top-grade performance without further degradation. This indicates that LECI has a stable training process with desired training dynamics; thus, it is practical to implement in real-world scenarios.

Dive into the training dynamics, during the initial phase of the training process depicted in Fig. 5, independence constraints are minimally applied (or are not applied), ensuring a conducive environment for discriminator training. This approach is predicated on the notion that adversarial gradients yield significance only post the successful training of discriminators according to Proposition 3.3. The observed performance "drop" is attributed to the fact that the generalization optimization is yet to be activated, so it indicates the general subgraph discovery network performance.

Figure 5: **Stability study.** The figures illustrate the OOD test accuracy curves during the training process on GOOD-Motif and GOOD-SST2. The big red arrows indicate the times when the independence constraints begin to take effect.

### 4.4 Ablation study

We empirically demonstrate the effectiveness of LECI through the above experiments. In this section, we further answer RQ5 to investigate the components of LECI. Specifically, we study the effect of environment adversarial (EA) training, label adversarial (LA) training, and pure feature shift consideration (PFSC) by attaching one of them to the basic interpretable subgraph discovery network. As shown in Tab. 3, it is clear that applying only partial independence obtains suboptimal performance. It may even lead to worse results as indicated by Motif-size. In comparison, much higher performance for the structure shift datasets can be achieved when

Table 3: **Ablation study on LECI.** The "None" and "Full" rows represent the results for the basic interpretable network and full LECI model N/A denotes that a certain component is not applied to the dataset.

|  | GOOD-Motif | | GOOD-CMNIST |
|---|---|---|---|
|  | basis | size | color |
| None | 58.38(9.52) | 65.17(6.48) | 33.41(4.63) |
| LA | 62.14(9.37) | 53.57(6.89) | 33.64(4.41) |
| EA | 64.02(21.30) | 38.69(1.86) | 38.29(9.85) |
| PFSC | N/A | N/A | 19.33(5.88) |
| Full | **84.56(2.22)** | **71.43(1.96)** | **51.80(2.70)** |

both EA and LA are applied simultaneously. This further highlights the importance of addressing the subgraph discovery challenges discussed in Sec. 3.2 to invariant predictions.

Overall, the experiments in this section demonstrate that LECI is a practical and effective method for handling out-of-distribution generalization in graph data. It outperforms existing baselines on both synthetic and real-world datasets and is robust to hyperparameter settings. Additionally, LECI's interpretable architecture and training stability further highlight its potential for real-world applications.

## 5 Conclusions & Discussions

We propose a technical and practical solution to incorporate two causal independence properties to release the potential of environment information for causal subgraph discovery in graph OOD generalization. The previous graph OOD works commonly assume the non-existence of the environment information, thus enabling these algorithms to work on more datasets without environment labels. However, the elimination of the environment information generally brings additional assumptions that may be more strict or even impossible to satisfy as mentioned in Sec. 3.7. In contrast, environment labels are widely used in the computer vision field [8]. While in graph learning areas, many labels can be accessed by applying simple groupings or deterministic algorithms as shown in GOOD [10] and DrugOOD [29]. These labels might not be accurate enough, but the experiments in Sec. 4 have proved their effectiveness over previous assumptions empirically. Moreover, a recent graph environment-aware non-Euclidean extrapolation work G-Splice [19] also validates the significance of environment labels from the augmentation aspect. Another avenue, graph environment inference [11, 12], has been explored deeper recently by GALA [20] which is conducive to LECI and future environment-based methods. We hope this work can shed light on the future direction of graph environment-centered methods.

## Acknowledgements

This work was supported in part by National Science Foundation grants IIS-2006861 and IIS-1908220.

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

# Appendix of LECI

## Contents

# A Broader Impacts

Out-of-distribution (OOD) generalization is a persistent challenge in real-world deployment scenarios, especially prevalent within the field of graph learning. This problem is heightened by the high costs and occasional infeasibility of conducting numerous scientific experiments. Specifically, in many real-world situations, data collection is limited to certain domains, yet there is a pressing need to generalize these findings to broader domains where executing experiments is challenging. By approaching the OOD generalization problem through a lens of causality, we open a pathway for integrating underlying physical mechanisms into Graph Neural Networks (GNNs). This approach harbors significant potential for wide-ranging social and scientific benefits.

Our work upholds ethical conduct and does not give rise to any ethical concerns. It neither involves human subjects nor introduces potential negative social impacts or issues related to privacy and fairness. We have not identified any potential for malicious or unintended uses of our research. Nevertheless, we recognize that all technological advancements carry inherent risks. Hence, we advocate for continual evaluation of the broader impacts of our methodology across diverse contexts.

# B Related works

## B.1 Extensive background discussions

**Out-of-Distribution (OOD) Generalization.** Out-of-Distribution (OOD) Generalization [39–42] addresses the challenge of adapting a model, trained on one distribution (source), to effectively process data from a potentially different distribution (target). It shares strong ties with various areas such as transfer learning [43–45], domain adaptation [46], domain generalization [25], causality [17, 18], invariant learning [9], and feature learning [47]. As a form of transfer learning, OOD generalization is especially challenging when the target distribution substantially differs from the source distribution. OOD generalization, also known as distribution or dataset shift [48, 49], encapsulates several concepts including covariate shift [50], concept shift [51], and prior shift [48]. Both Domain Adaptation (DA) and Domain Generalization (DG) can be viewed as specific instances of OOD, each with its own unique assumptions and challenges.

**Domain Adaptation (DA).** In DA scenarios [52–56], both labeled source samples and target domain samples are accessible. Depending on the availability of labeled target samples, DA can be categorized into supervised, semi-supervised, and unsupervised settings. Notably, unsupervised domain adaptation methods [8, 35, 57–71] have gained popularity as they only necessitate unlabeled target samples. Nonetheless, the requirement for pre-collected target domain samples is a significant drawback, as it may be impractical due to data privacy concerns or the necessity to retrain after collecting target samples in real-world scenarios.

**Domain Generalization (DG).** DG [25, 72–74] strives to predict samples from unseen domains without the need for pre-collected target samples, making it more practical than DA in many circumstances. However, generalizing without additional information is logically implausible, a conclusion also supported by the principles of causality [17, 18] (Appx. B.2). As a result, contemporary DG methods have proposed the use of domain partitions [8, 75] to generate models that are domain-invariant. Yet, due to the ambiguous definition of domain partitions, many DG methods lack robust theoretical underpinning.

**Causality & Invariant Learning.** Causality [17, 18, 21] and invariant learning [9, 24, 31, 76] provide a theoretical foundation for the above concepts, offering a framework to model various distribution shift scenarios as structural causal models (SCMs). SCMs, which bear resemblance to Bayesian networks [77], are underpinned by the assumption of independent causal mechanisms, a fundamental premise in causality. Intuitively, this supposition holds that causal correlations in SCMs are stable, independent mechanisms akin to unchanging physical laws, rendering these causal mechanisms generalizable. An assumption of a data-generating SCM equates to the presumption that data samples are generated through these universal mechanisms. Hence, constructing a model with generalization ability requires the model to approximate these invariant causal mechanisms. Given such a model, its performance is ensured when data obeys the underlying data generation assumption.

Peters et al. [21] initially proposed optimal predictors invariant across all environments (or interventions). Motivated by this work, Arjovsky et al. [9] proposed framing this invariant prediction

concept as an optimization process, considering one of the most popular data generation assumptions, PIIF. Consequently, numerous subsequent works [23, 24, 31, 78]—referred to as invariant learning—considered the initial intervention-based environments [21] as an environment variable in SCMs. When these environment variables are viewed as domain indicators, it becomes evident that this SCM also provides theoretical support for DG, thereby aligning many invariant works with the DG setting. Besides PIIF, many works have considered FIIF and anti-causal assumptions [24, 31, 78], which makes these assumptions popular basics of causal theoretical analyses. When we reconsider the definition of dataset shifts, we can also define the covariate data generation assumption, as illustrated in Fig. 1, where the covariate assumption is one of the most typical assumptions in DG [25].

**Graph OOD Generalization.** Extrapolating on non-Euclidean data has garnered increased attention, leading to a variety of applications [79–89]. Inspired by Xu et al. [90], Yang et al. [91] proposed that GNNs intrinsically possess superior generalization capability. Several prior works [92–94] explored graph generalization in terms of graph sizes, with Bevilacqua et al. [94] being the first to study this issue using causal models. Recently, causality modeling-based methods have been proposed for both graph-level tasks [4–7, 12] and node-level tasks [13]. However, except for CIGA [6], their data assumptions are less comprehensive compared to traditional OOD generalization. CIGA, while recognizing the importance of diverse data generation assumptions (SCMs), misses the significance of environment information and attempts to fill the gap through non-trivial extra assumptions, which we will discuss in Appx. B.2 and C.2.

Additionally, environment inference methods have gained traction in graph tasks, including EERM [13], MoleOOD [12], and GIL [11]. However, these methods face two undeniable challenges. First, their environment inference results require environment exploitation methods for evaluation, but there are no such methods that perform adequately on graph tasks according to the synthetic dataset results in GOOD benchmark [10]. Second, environment inference is essentially a process of injecting human assumptions to generate environment partitions, as we will explain in Appx. B.2, but these assumptions are not well compared. Hence, this paper can also be viewed as a work that aids in building better graph-specific environment exploitation methods for evaluating environment inference methods. Although we cannot directly compare with EERM and MoleOOD due to distinct settings, we strive to modify GIL and compare its environment exploitation method (IGA [15]) with ours under the same settings in Appx. B.3.

**Graph OOD Augmentation.** Except for representation learning, graph augmentation is also a promising way to boost the model's generalizability. While previous works focus on random or interpolation augmentations [95–97], Li et al. [19] recently explored environment-directed extrapolating graphs in both feature and non-Euclidean spaces, achieving significant results, which sets a solid foundation and renders a promising future data-centric direction, especially in the circumstance of the popularity of large language models.

**Relation between GNN Explainability/Interpretability and Graph OOD Generalization.** GNN explainability research [38, 98–101] aims to explain GNNs from the input space, whereas GNN interpretability research [4, 5, 102] seeks to build networks that are self-explanatory. Subgraph explanation [103, 104] is one of the most persuasive ways to explain GNNs. Consequently, subgraphs have increasingly become the most natural components to construct a graph data generation process, leading to subgraph-based graph data generation SCMs [6], inspired by general OOD SCMs [31].

**Comparisons to the previous graph OOD works.** It is worth mentioning that compared to previous works DIR [4] and GSAT [5], we do not propose a new subgraph selection architecture, instead, we propose a learning strategy that can apply to any subgraph selection architectures. While DIR defines their invariance principle with a necessity guarantee, they still cannot tackle the two subgraph discovery challenges (Appx. C.2) and only serve for the FIIF SCM. In contrast, we provide both sufficiency and necessity guarantees to solve the two subgraph discovery challenges under three data generation assumptions. We provide more detailed comparisons in terms of theory and assumptions in Appx. C.2.

## B.2 Environment information: significance and fairness discussion

In this section, we address three pivotal queries: **Q1:** Is it feasible to infer causal subgraphs $G_C$ devoid of environment information (partitions)? **Q2:** How do environment partitions generated by

environment inference methods [11–13, 28] compare to those chosen by humans? **Q3:** Are the experimental comparisons in this paper fair?

**Q1.** As depicted in Fig. 1, $G_S$ intrinsically correlates with multiple variables, namely, $G_S$, $Y$, and $G$. Similarly, $G_C$ has correlations with $G_S$, $Y$, and $G$. Absent an environment variable $E$, $G_C$, and $G_S$ can interchange without any repercussions, rendering them indistinguishable. Prior graph OOD methods have therefore necessitated supplementary assumptions to aid the identification of $G_C$ and $G_S$, such as the latent separability $H(G_C|Y) \leq H(G_S|Y)$ outlined by Chen et al. [6]. However, both $H(G_C|Y) \leq H(G_S|Y)$ and its converse $H(G_C|Y) \geq H(G_S|Y)$ [7] can be easily breached in real-world scenarios. Thus, without environment partitions, achieving OOD generalization is impracticable without additional assumptions.

Upon viewing environment partitions as a variable (Fig. 1), the environment variable effectively acts as an instrumental variable [17] for approximating the exclusive causation between $G_S$ and $G$. Consequently, we can discern the causal mechanism $G_S \rightarrow G$ through $E$, thereby distinguishing between $G_C$ and $G_S$. We discuss our proposed method and analysis, grounded in intuitive statistics and information theory, in our main paper's method section 3.

**Q2.** ZIN [16], a recent work, aptly addresses this question. Lin et al. [16] demonstrate that without predefined environment partitions, there can be multiple environment inference outcomes, with different outcomes implying unique causal correlations. This implies that absent prearranged environment partitions, causal correlations become unidentifiable, leading to conflicting environment inference solutions. Thus, guaranteeing environment inference theoretically is possible only via the use of other environment partition equivalent information or through the adoption of plausible assumptions. Therefore, these two factors, extra information, and assumptions, are intrinsically linked, as they serve as strategies for integrating human bias into data and models.

**Q3.** Building upon the last two question discussions, for all methods [5, 6, 11–13] that offer theoretical guarantees, they utilize different forms of human bias, namely, environment partitions or assumptions. Whilst numerous graph OOD methods compare their assumptions, our work demonstrates that simply annotated environment partitions outperform all current sophisticated human-made assumptions, countering Yang et al. [12]'s belief that environment information on graphs is noisy and difficult to utilize. Therefore, the comparisons with ASAP [34], DIR [4], GSAT [5], and CIGA [6] underscore this point. Additionally, through comparisons to conventional environment exploitation methods (IRM [9], VREx [14], etc.) in Sec. 4 and the comparison to the environment exploitation phase of GIL [11] (*i.e.*, IGA [15]) provided in Appx. B.3, we empirically validate that LECI currently surpasses all other environment exploitation methods in graph tasks. Therefore, our experiments justly validate these claims without engaging any fairness issues.

### B.3 Comparisons to previous environment-based graph OOD methods

In this section, we detail comparisons with prior environment-based graph OOD methods from both qualitative and quantitative perspectives. Firstly, we establish that we propose an innovative graph-specific environment exploitation approach. Secondly, we focus on comparing only the environment exploitation phase of previous work, discussing the methods chosen for comparison and the rationale.

#### B.3.1 Claim justifications: the innovative graph-specific environment exploitation method

**Motivations.** The motivation for our work stems from the fact that general environment exploitation methods have been found wanting in the context of graph-based cases, leading to our development of a graph-specific environment exploitation method. We demonstrate significant improvement with the introduction of this method. Subsequently, we justify LECI as an innovative graph-specific environment exploitation technique.

**Justifications.** An environment-based method usually comprises two phases: environment inference and environment exploitation. Environment inference aims to predict environment labels, while environment exploitation leverages these labels once they are available. The relationship between these two phases is not reciprocal; an environment inference method may employ an environment exploitation method to assess its performance, but the converse is not true.

Recent environment inference methods at the graph level [11, 12] and even the node level [13] do not incorporate graph-specific environment exploitation techniques. Our method, LECI, differs in this respect, as shown in the table below:

Table 4: Comparison of Characteristics to Previous Environment-Based Graph OOD Methods.

| Method | Environment Inference | Environment Exploit |
|---|---|---|
| MoleOOD [12] | Not Graph-Specific | Not Graph-Specific |
| GIL [11] | Graph-Specific | Not Graph-Specific |
| EERM [13] (Node-Level) | Graph-Specific | Not Graph-Specific |
| Ours (LECI) | N/A | Graph-Specific |

The criterion to determine if an environment exploitation method is graph-specific is whether the environment partition assistant is directly applied to graph-specific components (adjacency matrix: $A_C$, $A_S$) rather than common hidden representations.

This assessment can also be based on a replacement test: does the environment exploitation optimization remain valid after replacing $G$ and GNN with feature vectors $X$ and MLP?

As shown in Eq. 9, when we replace Equation (7) in MoleOOD [12], Equation (8) in GIL [11], and Equation (5) in EERM [13] with the following optimization objectives:

$$\text{MoleOOD: } \frac{1}{|\mathcal{X}|} \sum_{(X,y)\in\mathcal{X}} \left| \log q_\theta(y \mid X) - \mathbb{E}_{p(\mathbf{e}|\mathbf{X})}\left[\log p_\tau(y \mid X, e)\right] \right|$$

$$+ \beta\mathbb{E}_{\mathbf{e}}\left[ \frac{1}{|\mathcal{X}^e|} \sum_{(X,y)\in\mathcal{X}^e} \left[ -\log q_\theta(y \mid X) \right] \right] \tag{9a}$$

$$\text{GIL: } \mathbb{E}_{e\in\text{supp}(\mathcal{E}_{infer})} \mathcal{R}^e(f(\mathrm{X}), \mathrm{Y}; \theta) + \lambda \, \text{trace}\left( \text{Var}\, \mathcal{E}_{infer}\left(\nabla_\theta \mathcal{R}^e\right) \right) \tag{9b}$$

$$\text{EERM: } \min_\theta \text{Var}\left( \left\{ L\left(X^k, Y; \theta\right) : 1 \le k \le K \right\} \right) + \frac{\beta}{K} \sum_{k=1}^{K} L\left(X^k, Y; \theta\right) \tag{9c}$$

where $\theta$, $\tau$ are MLP parameters 9a; $f(\cdot)$ is MLP 9b; in Eq. 9c, $g_{w_k^*}(G)$ is replaced with $X^k$ (samples from environment $k$) since $g_{w_k^*}$ belongs to the environment inference phase. These three optimizations remain valid, thereby suggesting environment exploitation methods in [11–13] are not graph-specific. Note that this does not negate the graph-specific nature of their environment inference phases and network designs. For example, while GIL employs subgraph discovery networks as a basic generalization architecture, its invariant learning loss does not leverage this specialized architecture, as shown in Eq. 9b.

In contrast, substituting $G$ with $X$ in our environment optimization leads to problems. For instance, terms such as $\hat{X}_C$, and operations like $\hat{X}_C \subseteq X$ and $\hat{X}_S = X - \hat{X}_C$ would need redefinition; also, the mapping from $(X, A)_C$ to $X_C$ is not unique. Hence, this optimization is inherently graph-specific.

Therefore, LECI emerges as an innovative graph-specific environment exploitation method that offers clear differentiation from previous environment-based graph OOD methods. It could be considered appropriate to claim that LECI is the first of its kind in the realm of graph-specific environment exploitation.

### B.3.2 Environment exploitation quantitative comparisons

In this section, we compare to the environment-based graph OOD baseline GIL [11], since its original implementation is on a subgraph discovery network similar to the subgraph discovery network [5] we adopt. Since MoleOOD [12] has a design specific for molecules without using subgraph discovery networks, and EERM [13] is tackling node-level tasks, their settings are too different to be fairly compared with our method.

To conduct a fair comparison, we replace GIL's subgraph discovery architecture with GSAT [5] to ensure LECI and GIL are measured under the same network structure. Since GIL applies IGA [15]

as its invariant learning regularizer (environment exploitation method), we adopt the official IGA implementation from Domainbed [8]. The hyperparameter sweeping space is strictly aligned with the original GIL paper, *i.e.*, $\{10^{-1}, 10^{-3}, 10^{-5}\}$. After 3 runs with different random seeds, we have the following OOD validated test results on Motif-base, Motif-size, CMNIST, GOOD-SST2, GOOD-Twitter, GOOD-HIV-scaffold, GOOD-HIV-size, and DrugOOD-assay.

Table 5: **Environment exploitation phase comparisons: Sanity checks**

| Sanity checks | Motif-base | Motif-size | CMNIST |
|---|---|---|---|
| GIL (IGA) | 55.99(3.62) | 54.59(4.99) | 38.39(3.38) |
| LECI | **84.56(2.22)** | **71.43(1.96)** | **51.80(2.70)** |

Table 6: **Environment exploitation phase comparisons: Real-world**

| Real-world | GOOD-SST2 | GOOD-Twitter | GOOD-HIV-scaffold | GOOD-HIV-size | DrugOOD-assay |
|---|---|---|---|---|---|
| GIL (IGA) | 75.04(5.24) | **60.01(0.47)** | 71.27(1.69) | 62.27(0.91) | 73.20(0.12) |
| LECI | **83.44(0.27)** | 59.64(0.15) | **74.43(1.69)** | **65.44(1.78)** | **73.45(0.17)** |

The results indicate that except on Twitter, GIL [11] produces sub-optimal results compared to LECI.

## C  Theory and discussions

In this section, we will first provide theoretical proofs of the relaxed independence property, Proposition 3.3, Lemma 3.4, Lemma 3.5, Lemma 3.6, Theorem 3.7, and Theorem 3.8. Then, we provide specific challenge-solving, assumption comparisons for previous graph OOD methods. Subsequently, we will discuss the subgraph discovery challenges and the subgraph selector invariance. Finally, we will provide the current challenges, limitations, possible solutions, and future directions under the subgraph modeling data generation assumptions.

### C.1  Proofs

#### C.1.1  Relaxed independence property

**Proposition C.1.** *For any $G_p$ that $G_S \subseteq G_p$, $I(G_S; Y) \le I(G_p; Y)$.*

*Proof.* $I(G_S; Y) \le I(G_p; Y)$ is equivalent to $H(Y|G_S) \ge H(Y|G_p)$ where $H$ denotes entropy, since $H(Y)$ is a constant. We consider a random graph $G_S$ consisting of a set of random variables (edges) $A_S$. $G_S \subseteq G_p$ means $G_p$'s set of random edges $A_p$ is a superset of $A_S$, *i.e.*, $A_p = A_S \cup A_+$, where $A_+$ is the set of additional edges in $A_p$. Therefore, $H(Y|G_S) = H(Y|A_S) \ge H(Y|A_S, A_+) = H(Y|A_p) = H(Y|G_p)$, *i.e.*, $I(G_S; Y) \le I(G_p; Y)$. □

#### C.1.2  Proof of Proposition 3.3

**Proposition C.2.** *Denoting KL-divergence as $KL[\cdot\|\cdot]$, for $\theta$ fixed, the optimal discriminator $\phi_E$ is $\phi_E^*$, s.t.*

$$KL\left[P(E|\hat{G}_C)\|P_{\phi_E^*}(E|\hat{G}_C)\right] = 0. \tag{10}$$

*Proof.* Since given a fixed $\theta$, $I(E; \hat{G}_C)$ and $H(E)$ are both constant. Therefore, we have

$$\begin{aligned}
\phi_E^* &= \operatorname{argmin}_{\phi_E} - \mathbb{E}\left[\log P_{\phi_E}(E|\hat{G}_C)\right] \\
&= \operatorname{argmin}_{\phi_E} I(E; \hat{G}_C) - \mathbb{E}\left[\log P_{\phi_E}(E|\hat{G}_C)\right] - H(E) \tag{11} \\
&= \operatorname{argmin}_{\phi_E} \mathrm{KL}\left[P(E|\hat{G}_C)\|P_{\phi_E}(E|\hat{G}_C)\right].
\end{aligned}$$

□

### C.1.3 Proof of Lemma 3.4

**Lemma C.3.** *Given a subgraph $G_p$ of input graph $G$ and SCMs (Fig. 1), it follows that $G_p \subseteq G_C$ if and only if $G_p \perp\!\!\!\perp E$.*

*Proof.* ($\Rightarrow$) Contradiction: If $G_p \subseteq G_C$, but $G_p$ is not independent to $E$, i.e., $I(E; G_p) > 0$. Since $I(E; G_C) \geq I(E; G_p)$ (this is trivial because $G_p$ is a subset of $G_C$), we obtain $I(E; G_C) > 0$, which contradicts to the fact that $G_C \perp\!\!\!\perp E$.

($\Leftarrow$) Contradiction: If $G_p \perp\!\!\!\perp E$ but $G_p$ is not a subgraph of $G_C$; i.e., $\exists$ non-trivial (not empty) $G_q \subseteq G_p$, s.t. $G_q \subseteq G_S$. According to the three SCMs, $G_q$ is not independent of $E$; thus, $G_p$ is not independent of $E$, which contradicts the premise.

Since $\text{KL}\left[P(E|\hat{G}_C)\|P_{\phi_E}(E|\hat{G}_C)\right] \geq 0$, when $\phi_E = \phi_E^*$, $\text{KL}\left[P(E|\hat{G}_C)\|P_{\phi_E}(E|\hat{G}_C)\right]$ reaches its minimum 0. $\qquad\square$

### C.1.4 Proof of Lemma 3.5

**Lemma C.4.** *Given SCMs (Fig. 1), the predicted $\hat{G}_C = f_\theta(G)$, and the EA training criterion (Eq. 6), $\hat{G}_C \subseteq G_C$ if and only if $\theta \in \Theta_E^*$.*

*Proof.* $\theta \in \Theta_E^*$ is equivalent to $\theta = \arg\min_\theta I(E; \hat{G}_C)$, which indicates $I(E; \hat{G}_C) = 0$ when $\theta \in \Theta_E^*$. Since according to Lemma 3.4, we have $\hat{G}_C \subseteq G_C$ if and only if $\hat{G}_C \perp\!\!\!\perp E$, while $I(E; \hat{G}_C) = 0$ if and only if $\hat{G}_C \perp\!\!\!\perp E$, it follows that $\hat{G}_C \subseteq G_C$ if and only if $\theta \in \Theta_E^*$. $\qquad\square$

### C.1.5 Proof of Theorem 3.7

**Theorem C.5.** *Under the covariate SCM assumption and both EA and LA training criteria (Eq. 6 and 7), it follows that $\hat{G}_C = G_C$ if and only if $\theta \in \Theta_E^* \cap \Theta_L^*$.*

*Proof.* Under the covariate SCM assumption, Lemma 3.6 is satisfied. According to Lemma 3.5 and 3.6, EA and LA are both optimized ($\theta \in \Theta_E^* \cap \Theta_L^*$) if and only if $\hat{G}_C \subseteq G_C$ and $\hat{G}_S \subseteq G_S$. Since $\hat{G}_C = G - \hat{G}_S$ and $G_C = G - G_S$, $\hat{G}_S \subseteq G_S$ is equivalent to $G_C \subseteq \hat{G}_C$. These two inclusive relations $G_C \subseteq \hat{G}_C$ and $\hat{G}_C \subseteq G_C$ imply $\hat{G}_C = G_C$. Therefore, we obtain our conclusion that $\hat{G}_C = G_C$ if and only if EA and LA both reach their optimums. $\qquad\square$

### C.1.6 Proof of Theorem 3.8

**Theorem C.6.** *Given the covariate/FIIF/PIIF SCM assumptions and both EA and LA training criteria (Eq. 6 and 7), it follows that $G_C = \hat{G}_C$ if and only if, under the optimal EA training, the LA criterion is optimized, i.e., $\theta \in \arg\max_{\theta \in \Theta_E^*}\{\min_{\phi_L} \mathcal{L}_L\}$.*

*Proof.* According to Lemma 6, we have $\hat{G}_C \perp\!\!\!\perp E$ under the optimal EA training. Recall the relaxed property: for any $G_p$ that $G_S \subseteq G_p$, $I(G_S; Y) \leq I(G_p; Y)$, which is valid under the three SCMs 1. Under the premise of $\hat{G}_C \perp\!\!\!\perp E$ that is equivalent to $G_S \subseteq \hat{G}_S$, we take the relaxed property into account, *i.e.*, for $\hat{G}_S$ that $G_S \subseteq \hat{G}_S$, we have $I(G_S; Y) \leq I(G_p; Y)$. Therefore, when we apply the LA training criterion under the optimal EA training, the mutual information $I(\hat{G}_S; Y)$ can be minimized to be $\hat{G}_S = G_S$, *i.e.*, $\hat{G}_C = G_C$. $\qquad\square$

### C.2 Theory and assumption comparisons with previous graph OOD methods

Many recent graph OOD learning strategies [4–6] adopt interpretable subgraph selection architectures, but our method is distinct from them in terms of the data generation assumptions, motivated challenges, and invariant learning strategy. To further investigate the key reason behind the LECI's OOD generalization ability, in Tab. 7, we provide a clear comparison with recent OOD generalization methods *w.r.t.* the subgraph discovery challenges introduced in Sec. 3.2. Specifically, DIR [4] proposes to learn causal subgraphs with adaptive interventions. However, its architecture requires the

Table 7: **Comparison of related methods on addressing the two challenges.** We compare the problem-solving capabilities of our proposed method, LECI, with prior methods and the two components of LECI. In the table, check marks (✓) denote that the method is capable of addressing the challenge. Cross marks (✗) indicate that the method is unable to solve the problem, or the preconditions are nearly impossible to be satisfied. Exclamation marks (!) imply that the method can alleviate the problem under restricted assumptions.

| Method | DIR | GSAT | CIGAv1 | CIGAv2 | $E \perp\!\!\!\perp G_C$ | $Y \perp\!\!\!\perp G_S$ | LECI |
|---|---|---|---|---|---|---|---|
| Precision 1 | ✗ | ✗ | ✗ | ! | ✓ | ✗ | ✓ |
| Precision 2 | ✗ | ✗ | ✗ | ! | ✗ | ✓ | ✓ |

pre-selected size information of the true causal subgraphs. It proposes to solve subgraph discovery challenges under the FIIF assumption, but it fails to provide a sufficient condition of its theoretical guarantees. GSAT [5] proposes to use information constraints to select causal subgraphs under only the FIIF assumption. Its invariant guarantee relies on the anti-causal correlation $Y \rightarrow G_C$, *i.e.*, given $Y$, $G_C$ should be unique, which does not hold in the general scenario where each class can have multiple modes. CIGA [6] is the first graph OOD method that can work on both FIIF and PIIF assumptions. However, CIGAv1 admits the limitation of the requirement of the size of true causal subgraphs, which is not practical even with human resources. CIGAv2 tries to achieve invariant predictions without the requirement of the true size information, but it, instead, requires an implicit assumption of the independence between the selected spurious subgraphs and the non-selected spurious subgraphs (denoted as $\widehat{G}_s^p$ and $\widehat{G}_s^l$ in CIGA), according to the equation (34) in its appendix. This assumption, named as independent components assumption, though weaker than the requirement of CIGAv1, still limits its applications, because $E$, as the confounder of all elements in $G_S$, can destroy this independence from a finer modeling perspective. Additionally, except for the data generation assumptions (FIIF and PIIF), it requires an invariant feature assumption $H(G_C|Y) \leq H(G_S|Y)$. However, both $H(G_C|Y) \leq H(G_S|Y)$ and its converse $H(G_C|Y) \geq H(G_S|Y)$ [7] can be easily breached in real-world scenarios. Note that, for simplicity, under the subgraph-based data generation assumptions, we slightly abuse $G_C$ and $G_S$ as the latent causal and non-causal factors $C$ and $S$.

Learning from the limitations of previous works, LECI is an innovative and effective method that addresses the problem of OOD generalization in graphs by exploiting environment information and learning causal independence. Specifically, LECI addresses the two subgraph discovery challenges by introducing two independence properties as $E \perp\!\!\!\perp G_C$ and $Y \perp\!\!\!\perp G_S$. By simultaneously addressing these challenges, LECI is able to alleviate the two precision problems under the three data generation assumptions. This makes LECI a unique and promising approach for OOD generalization in graph data.

**Assumption comparisons.** As we have compared previous works on subgraph discovery ability, to avoid possible confusion, we further provide a clear comparison in Tab. 8 with respect to data generation assumptions (SCMs), theory, and theoretical guarantees. We would like to argue that simple annotated/grouped environment partitions are strong enough to outperform many sophisticated human-designed assumptions.

Table 8: **Assumption comparisons.**

| Methods | SCMs | Theory | Guarantees | Extra information (assumptions) |
|---|---|---|---|---|
| DIR | FIIF | $Y \perp\!\!\!\perp G_S|G_C$ | Necessity | True subgraph size assumption (TSSA) |
| GSAT | FIIF | Information bottleneck | Sufficiency & Necessity | Reversible predictor assumption |
| CIGAv1 | FIIF & PIIF | Intra-class similarity | Sufficiency & Necessity | TSSA |
| CIGAv2 | FIIF & PIIF | Intra-class similarity | Sufficiency & Necessity | $H(G_C|Y) \leq H(G_S|Y)$ & independent components assumption |
| LECI | FIIF & PIIF | Two independence & relaxed property | Sufficiency & Necessity | Environment partitions |

## C.3 Two precision challenges discussion

Under the subgraph-based graph modeling, it is obvious that the two precision challenges should be solved simultaneously to achieve invariant GNNs; thus, we argue that these two challenges should be analyzed and compared explicitly. While many previous works rely on unilateral causal analysis without discussing and comparing assumptions explicitly, we strive to compare them in the last subsection.

In this section, we would like to share a failure case to discuss the reasons behind the difficulty of solving the two subgraph discovery challenges. To be more specific, the difficulty may hide in the design of subgraph discovery architecture, *e.g.*, maximizing mutual information $I(Y; \hat{G}_C)$ can be much less effective than expected. Generally, it is reasonable to believe that the invariant predictor $g_{\mathrm{inv}}$ in Fig. 2 is able to solve the second precision challenge by maximizing the mutual information between the selected subgraph and the label, *i.e.*, $I(Y; \hat{G}_C)$. This is because maximizing $I(Y; \hat{G}_C)$ is approximately equivalent to minimizing $I(Y; \hat{G}_S)$, *i.e.*, enforcing $\hat{G}_S$ to be independent of $Y$. This belief, however, does not hold empirically according to the ablation study in Sec. 4.4, where the subgraph discovery network with only the solution of the first precision problem (EA) fails to generalize. Intuitively, the subgraph selector $f_\theta$ is differentiable, but due to its discrete sampling nature, there are many undefined continual states, *e.g.*, the middle states of the existence of an edge. These undefined states will hinder the effectiveness of gradients propagated to the subgraph selector. As a result, the invariant predictor can hardly implicitly access the label-correlated information from the unselected part $\hat{G}_S$ through gradients. This implies that training the invariant predictor alone is not sufficient to address the second precision challenge without support from the $\hat{G}_S$ side. Therefore, to explain LECI's success intuitively, we can consider $g_{\mathrm{inv}}$ and $g_{\mathrm{L}}$ as the two information probes from both $\hat{G}_C$ and $\hat{G}_S$ sides so that we exploit information from both sides to fully support maximizing $I(Y; \hat{G}_C)$.

From the perspective of the LECI optimization, each subgraph discovery challenge has verifiable and unverifiable optimization targets. The unverifiable descriptions include (1) $\hat{G}_S$ *fails to cover* $G_S$; (2) $\hat{G}_C$ *fails to cover* $G_C$. These unverifiable descriptions cannot be formed as optimization objects because it is impossible to verify whether one can cover all information unless one enumerates the whole search space. Therefore, we adopt the verifiable objectives (1) $\hat{G}_C$ *contains parts of* $G_S$ and (2) $\hat{G}_S$ *contains parts of* $G_C$.

### C.4 How can the subgraph selector keep invariant under distribution shifts?

We discuss this question from two aspects: (1) Can subgraph selector $f_\theta$ recognize $G_C$ with unseen $G_S$? (2) Will subgraph selector $f_\theta$ be misled to recognize parts of unseen $G_S$ as $G_C$?

This discussion is under the following assumptions and conditions:

- **Causal subgraph equal support assumption (CSES):** Both $P^{tr}(G_C)$ and $P^{te}(G_C)$ have the same support, where $P(G_C) > 0$.

- **Architecture:** Assume $f_\theta$ is a $K$-layer message passing neural network that only captures $K$-hop subgraph information. Theoretically, we have an optimal injective $K$-hop subgraph classifier that can distinguish all $K$-hop subgraphs. This implies only the same $K$-hop subgraphs can produce the same result by this classifier, while similar subgraphs cannot.

- **Information sufficiency:** Assume $K$ be certain value that $K$-hop subgraphs are large enough to cover $G_C$.

(1) In essence, $f_\theta$ operates as a classifier. When provided with an edge, it classifies the associated ego-graph of the edge into either label 1 (when the central edge is part of $G_C$) or label 0 (otherwise). We denote the distribution of $K$-hop ego-graphs on the edges of $G_C$ as $P(G_C^K)$, and the $K$-hop ego-graph space as $\mathcal{G}^K$. Therefore, we have $f_\theta : \mathcal{G}^K \mapsto \{0, 1\}$, where $\forall G_p^K \in \mathcal{G}^K, f_\theta(G_p^K) = 1$ when and only when $P(G_C^K = G_p^K) > 0$, i.e., the distribution of $f_\theta$ is only affected by the support of $P(G_C^K)$. The CSES assumption can be relaxed to that $P^{tr}(G_C^K)$ and $P^{te}(G_C^K)$ have the same support. This ensures that $G_C$ with unseen $G_S$ can still be recognized. Note that when $K$ is smaller, the assumption is looser. In contrast, when $K \to +\infty$, this assumption is equivalent to that $P^{tr}(G)$ and $P^{te}(G)$ have the same support, which is stringent. Therefore, $K$ is the "locality rate" for us to loosen a global equal support requirement to a local one.

(2) To answer the second question, we will prove the following proposition:

**Proposition C.7.** *There is not a $K$-hop subgraph that shares across $P(G_C)$ and $P^{te}(G_S)$.*

*Proof.* Contradiction:

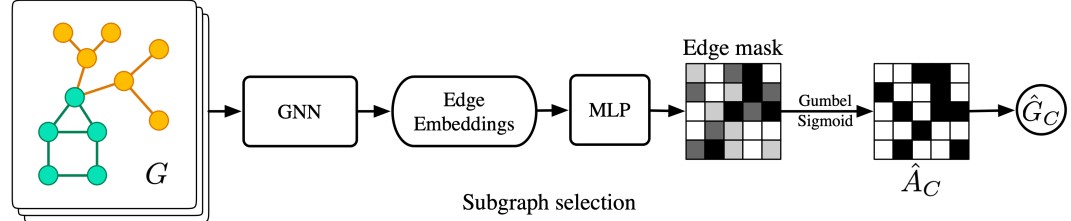

Figure 6: **The architecture of subgraph selector $f_\theta$.**

Given the aforementioned assumptions, suppose there is a $K$-hop subgraph that shares across $P(G_C)$ and $P^{te}(G_S)$ as described in your example. This indicates that $\exists G_p^K \subseteq P^{tr}(G_C^K)$ such that there is an edge $e_s$ in $P^{te}(G_S)$ whose corresponding $K$-hop subgraph $G_{e_s}^K$ (centered with this edge) equals to $G_p^K$, i.e., $G_{e_s}^K = G_p^K$. Since $G_p^K \subseteq P^{tr}(G_C^K)$, the central edge $p$ is an edge on $G_C$, and $G_C \subseteq G_p^K$ that is implied by the first condition above. Since $G_{e_s}^K = G_p^K$, the edge $e_s$ is an edge of $G_C$, and $G_C \subseteq G_{e_s}^K$. Therefore, since our basic assumption indicates an edge can either belong to $G_C$ or $G_S$, and $e_s$ belongs to $G_C$, $e_s$ cannot belong to $G_S$ even though $G_S \in P^{te}(G_S)$, which contradicts the supposition. Therefore, there is no such $K$-hop subgraph that shares across $P(G_C)$ and $P^{te}(G_S)$. $\square$

In conclusion, under these three assumptions, it can be argued that $P(G_C|G)$ remains invariant when such $f_\theta$ is employed.

### C.5    Challenges and Limitations

This section discusses the potential challenges, limitations of our methodology, and future research directions that can be pursued to further refine our approach.

A significant constraint in our method arises from the utilization of common adversarial training. This technique often requires longer training times due to the need for discriminator training. The attainment of optimal discriminators becomes particularly challenging when subgraph distributions continue to shift as a result of the subgraph selection training. A potential area for future research could be the exploration of alternative ways to enforce independence properties, for instance, approximations of the min-max optimization could be employed.

Another potential drawback of our method pertains to the subgraph discovery architecture [5] that we use. As previously discussed in Section C.3, separating causal from non-causal subgraphs complicates the training process. In response, additional information from both subgraph branches must be provided to facilitate comprehensive optimizations. Consequently, future research could involve the development of new, promising subgraph discovery architectures that alleviate these challenges.

Our method might not directly correlate with OOD errors. For context, current statistical graph modeling techniques, such as [94, 105, 106], can relate methods to OOD generalization errors, albeit focusing on size shift analysis. Methods inspired by explanations, like DIR [4] and our proposed approach, offer practical solutions capable of addressing a wider range of shifts. Nonetheless, bridging these methods to OOD generalization errors presents a substantial challenge.

The limitations and future directions discussed herein have been outlined within the confines of the current three data generation assumptions. Moving forward, future studies could consider a wider range of data generation assumptions. Viewing from the lens of graph explainability, these assumptions could not only be at the subgraph-level but could also explore edge-level, node-level, or even flow-level [101] considerations. Such explorations would be instrumental in contributing to the continuous evolution and refinement of graph OOD models.

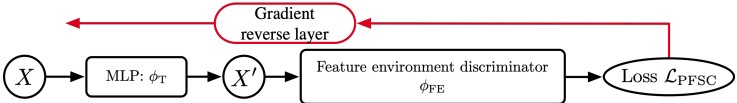

Figure 7: **The architecture of the pure feature shift consideration (PFSC).** The figure illustrates the overall network of the PFSC. The reverse gradient multiplication hyperparameter is denoted as $\lambda_{\text{PFSC}}$. The loss $\mathcal{L}_{\text{PFSC}}$ here is the negative log-likelihood in Eq. 12.

## D    LECI implementation details

### D.1    The architecture of subgraph selector

The architecture of our subgraph selector, as illustrated in Fig. 6, is based on the work of Miao et al. [5] and uses edges to select subgraphs. The input graph $G$ is passed through a general GNN to obtain the hidden representations of its nodes. These representations are then transformed into edge embeddings, where each edge embedding is the concatenation of the corresponding node representations. To obtain the edge probability mask, we use a multi-layer perceptron (MLP) and a sigmoid function to estimate the sampling probability of each edge. With these sampling probabilities, we obtain the sampled adjacency matrix $\hat{A}_C$, where we apply the Gumbel-sigmoid [32] technique to make the network differentiable. Finally, the subgraph selector outputs the selected subgraph $\hat{G}_C$ composed of $\hat{A}_C$ and the corresponding remaining nodes. Practically, the $\hat{G}_C$ composes all nodes but edge weights are assigned as $\hat{A}_C$, which is the typical edge representation of subgraphs. Note that to avoid trapping into local minimums caused by artifacts in synthetic datasets, we apply temperature annealing from 10 to 0.1 for Gumbel-sigmoid.

### D.2    The architecture of the pure feature shift consideration

The technique of pure feature shift consideration (PFSC) is used to remove environment information from the node feature $X$. As illustrated in Fig. 7, a small MLP transformer, denoted as $\text{MLP}_{\text{T}}$, with learnable parameters $\phi_{\text{T}}$ is used to transform the node features $X$ into environment-free node features $X'$. This transformation is achieved through an adversarial training process, where the MLP is trained to remove environment information from $X$, and the feature environment discriminator aims to approximate the correlation between $X'$ and the environment $E$. The objective of this process can be written as Eq. 12. After this transformation, the input graph $G$ is composed of $A$ and $X'$, *i.e.*, $G = (X', A)$. It is worth noting that the use of PFSC alone is not sufficient to achieve optimal performance, as shown in the results of the ablation study in Tab. 3.

$$\text{PFSC} : \max_{\phi_{\text{T}}} \min_{\phi_{\text{FE}}} -\mathbb{E}\left[\log P_{\phi_{\text{FE}}}(E|X')\right], \tag{12}$$

### D.3    LECI's training strategy

The training strategy of LECI encompasses two implementations. The first one adheres to a rigorous training process detailed in Sec. 3.4. In this approach, updates to the subgraph selector are performed only upon the convergence of the two discriminators. This iterative training procedure involving the subgraph selector and the two discriminators continues throughout the training process. Despite its thoroughness, this approach can be somewhat time-consuming.

To overcome this hurdle, we resort to the second implementation. Here, we optimize the entire network, including the subgraph selector and the discriminators. The adversarial training hyperparameters, $\lambda_{\text{L}}$ and $\lambda_{\text{E}}$, are initially assigned a value of zero. This strategy restricts the influence of the gradients from the discriminators on the subgraph selector, thereby ensuring a relatively stable distribution of $\hat{G}_C$ and consistent training of the discriminators.

As the training proceeds and network parameters stabilize over time, with the discriminators moving closer to convergence, we progressively increase the adversarial training hyperparameters. This strategy allows for a gradual implementation of the independence constraints, ensuring the discriminators remain near their optimal states without being abruptly disrupted, as depicted in Fig. 5.

# E  Detailed experiment settings

In this section, we provide experimental details including fairness discussions, datasets, baseline selection justifications, training and optimization settings, hyperparameter settings, and software/hardware environments.

## E.1  Fairness discussion

In all experiments in this paper, all reported results are obtained from 3 runs and are selected by automatic hyperparameter sweeping with respect to ID and OOD validation results. We also provide a fairness discussion about the usage of environment information in Appx. B.2.

## E.2  Datasets

For our experimental section, we utilize the SST2, Motif, and CMNIST datasets from the GOOD benchmark [10], along with the LBAP-core-ic50 assay split from DrugOOD [29]. Furthermore, we create the GOOD-Twitter and CFP-Motif datasets. All dataset splits target domain generalization and aligns with the guidelines provided in the original literature except CFP-Motif which incorporates PIIF and FIIF shifts as described in Sec. 4.2. Although the GOOD datasets are primarily designed for covariate splits, we acknowledge that these splits can include extra spurious correlations, meaning they may also partially satisfy the FIIF and PIIF assumptions.

**GOOD-HIV** is a compact, real-world molecular dataset, derived from MoleculeNet [107]. It contains molecular graphs in which atoms are nodes and chemical bonds are edges, and the task is to predict a molecule's potential to inhibit HIV replication. Gui et al. [10] developed splits based on two domain attributes: the Bemis-Murcko scaffold [108], and the number of nodes in a molecular graph.

**GOOD-SST2** is a sentiment analysis dataset based on real-world natural language, adapted from the work of Yuan et al. [38]. In this dataset, each sentence is converted into a grammar tree graph where individual words serve as nodes, with their associated word embeddings acting as node features. The main task within this dataset is a binary classification exercise aiming to predict the sentiment polarity of each sentence. Sentence lengths are selected as the domains, bringing an added layer of complexity to the classification task.

**GOOD-Twitter**, similarly, is a real-world natural language sentiment analysis dataset, adapted from Yuan et al. [38]. This dataset applies the same transformation process as in SST2, where sentences are converted into grammar tree graphs, with nodes representing words and node features derived from corresponding word embeddings. However, the classification task in this dataset is three-fold, tasked with predicting one of three sentiment polarities for each sentence. As with the GOOD-SST2 dataset, sentence lengths are selected as the domains.

**GOOD-CMNIST** is a semi-synthetic dataset specifically designed to test node feature shifts. It consists of graphs built from hand-written digits extracted from the MNIST database, utilizing superpixel techniques [37] for the transformation. Taking a page from the book of Arjovsky et al. [9], Gui et al. [10] assigned colors to digits based on their domains and concepts. More specifically, in the covariate shift split, digits are colored using seven distinct colors, with the digits exhibiting the first five colors allocated to the training set, those with the sixth color to the validation set, and the remaining digits with the seventh color allocated to the test set.

**GOOD-Motif** is a synthetic dataset, inspired by Spurious-Motif [4], created for structure shift studies. Each graph in the dataset is generated by connecting a base graph and a motif, with the motif solely determining the label. The base graph type and the size are chosen as domain features for creating domain generation splits. Specifically, they generate graphs using five label-irrelevant base graphs (wheel, tree, ladder, star, and path) and three label-determining motifs (house, cycle, and crane).

**Covariate/FIIF/PIIF Motif (CFP-Motif)** is a synthetic dataset similar to GOOD-Motif and is designed for further structure shift study. The first difference between GOOD-Motif and CFP-Motif is the base subgraphs of the OOD validation and test set. Specifically, instead of using paths as base subgraphs in the test set, we produce Dorogovtsev-Mendes graphs [36] as base subgraphs. While in the validation set, we assign base graphs as circular ladders. Second, CFP-Motif extends GOOD-Motif with FIIF and PIIF shifts. In FIIF and PIIF splits (Fig. 1), $G_C$ and $Y$ have a probability of 0.9 to determine the size of $G_S$, leading to spurious correlations w.r.t. size. That is after determining $G_C$

Table 9: Numbers of graphs in training, ID validation, ID test, OOD validation, and OOD test sets for the 7 datasets.

| Dataset | Train | ID validation | ID test | OOD validation | OOD test | Train | OOD validation | ID validation | ID test | OOD test |
|---|---|---|---|---|---|---|---|---|---|---|
| | | | Scaffold | | | | | Size | | |
| GOOD-HIV | 24682 | 4112 | 4112 | 4113 | 4108 | 26169 | 4112 | 4112 | 2773 | 3961 |
| | | | Length | | | | | | | |
| GOOD-SST2 | 24744 | 5301 | 5301 | 17206 | 17490 | | | | | |
| | | | Length | | | | | | | |
| GOOD-Twitter | 2590 | 554 | 554 | 1785 | 1457 | | | | | |
| | | | Color | | | | | | | |
| GOOD-CMNIST | 42000 | 7000 | 7000 | 7000 | 7000 | | | | | |
| | | | Basis/Size | | | | | | | |
| GOOD-Motif | 18000 | 3000 | 3000 | 3000 | 3000 | | | | | |
| | | Covariate/FIIF/PIIF | | | | | | | | |
| CFP-Motif | 1800 | 300 | 300 | 300 | 300 | | | | | |
| | | | Assay | | | | | | | |
| LBAP-core-ic50 | 34179 | 11314 | 11683 | 19028 | 19032 | | | | | |

and $Y$, the types of causal graphs (FIIF) and the labels (PIIF) will be used to choose the size of $G_S$ from [10, 20, 30] with variances. Note that FIIF and PIIF also contain base graph shifts.

**LBAP-core-ic50**, a dataset derived from DrugOOD [29], is applied in the task of Ligand-based affinity prediction (LBAP) with the core noise level and IC50 measurement type as domain features.

The selection of these datasets was deliberate to cover: (1) typical molecular predictions with three diverse domains: scaffold, size, and assay; (2) conventional natural language transformed datasets, SST2 and Twitter, with sentence lengths as domains; (3) synthetic datasets that encompass both structure shifts and feature shifts.

All GOOD datasets and datasets created with the GOOD strategy use the MIT license. DrugOOD dataset uses GPL3.0 license.

We provide the dataset statistics in Tab. 9.

### E.3  Baseline selection justification and discussion

This section offers an in-depth justification for our selection of experimental baselines. Traditional Out-Of-Distribution (OOD) baselines tend to perform comparably on graph tasks. As such, we elected to use the most typical ones as our baselines, including IRM [9], VREx [14], Coral [35], and DANN [8]. Specifically, DANN serves as an essential baseline to empirically demonstrate that direct domain adversarial training is ineffective for graph tasks. We have chosen not to compare our method with GroupDRO [22], as it is empirically similar to VREx. Furthermore, baselines such as EIIL [28], IB-IRM [31], and CNC [109] were deemed non-comparative due to their performances falling below that of the state-of-the-art graph-specific OOD method [6].

Graph-specific baselines largely concentrate on learning strategies within the subgraph discovery architecture, except for ASAP [34]. ASAP acts as the subgraph discovery baseline, employing intuitive techniques to select subgraphs. DIR [4] is the original graph-specific method for invariant learning, while GSAT [5] incorporates the information bottleneck technique [110, 111]. GIB [112], the first graph information bottleneck method, is not included in our experiments due to its performance is lagging behind other baselines, as demonstrated by Chen et al. [6]. CIGA [6], one of the most recent state-of-the-art methods, provides a thorough theoretical discussion about graph OOD generalization. Among these methods, DIR and GSAT perform similarly, with CIGA being the only method that clearly outperforms both ERM and ASAP in real-world scenarios, as seen in Fig. 2. The performance shortcomings of DIR and CIGA in synthetic datasets can be partially attributed to the selection of the subgraph size ratio.

In addition to CIGA, recent graph-specific baselines such as DisC [7], MoleOOD [12], and GIL [11] exist. We provide a comparison with GIL in Appx. B.3 concerning the environment exploitation phase. As for DisC and MoleOOD, we opt not to include them in our baseline due to specific reasons. DisC is not a learning strategy that can be universally applied to any subgraph discovery network, and its focus on data generation assumptions differs from ours. As for MoleOOD, it is

specific to molecule predictions, and a fair comparison with its environment exploration phase is not feasible, given its inability to work within a subgraph discovery architecture. A comparison with the entirety of MoleOOD would not provide a controlled comparison, as it would be unclear whether any improvements stem from the environment inference phase or the environment exploitation phase.

## E.4 Training settings

Among the variety of Graph Neural Networks (GNNs) [113–121], we elect to use the canonical three-layer Graph Isomorphism Network (GIN) [115] with a hidden dimensionality of 300, a dropout ratio of 0.5, and mean global pooling as the encoder for the GOOD-Motif basis split. For the GOOD-Motif size, CFP-Motif, GOOD-CMNIST, and real-world datasets, we extend GIN with the addition of virtual nodes. A linear classifier is used following the graph encoder. The reason for using virtual nodes for GOOD-Motif size and CFP-Motif is that these splits include size shifts which are hard to capture by mean pooling. Instead of using other pooling methods, we simply apply a virtual node to capture this global information.

Throughout the training phase, we utilize the Adam optimizer [122]. For GOOD-Motif and GOOD-SST2, the learning rate is set to $10^{-3}$. For CFP-Motif, the learning rate is set to $10^{-4}$ with a weight decay from $[0, 10^{-4}]$. For GOOD-HIV, GOOD-Twitter, and DrugOOD, we apply a learning rate of $10^{-4}$, accompanied by a weight decay of $10^{-4}$. Across all datasets, the standard number of training epochs is set to 200. A training batch size of 32 is used for all datasets, with the exception of GOOD-CMNIST, where we use a batch size of either 64 or 128.

## E.5 Hyperparameter sweeping

In this section, we display the hyperparameter sweeping space of each method. For the traditional OOD method with only one hyperparameter, we denote the only hyperparameter as $\lambda$. For methods ASAP, DIR, and CIGA, we denote the size ratio of the causal subgraphs compared to original graphs as $\lambda_s$.

### E.5.1 Hyperparameters of baselines

- **IRM:** $\lambda = [10^{-1}, 1, 10^1, 10^2]$.
- **VREx:** $\lambda = [1, 10^1, 10^2, 10^3]$.
- **Coral:** $\lambda = [10^{-3}, 10^{-2}, 10^{-1}, 1]$.
- **DANN:** $\lambda = [10^{-3}, 10^{-2}, 10^{-1}, 1]$.
- **ASAP:** $\lambda_s = [0.2, 0.4, 0.6, 0.8]$.
- **DIR:** $\lambda_s = [0.2, 0.4, 0.6, 0.8]$ and $\lambda_{var} = [10^{-2}, 10^{-1}, 1, 10^1]$, where $\lambda_{var}$ is the variance constraint.
- **GSAT:** $r = [0.5, 0.7]$ and $\lambda_{decay} = [10, 20]$. Here, $r$ and $\lambda_{decay}$ are the information constraint hyperparameter and the information constraint 0.1 decay epochs, respectively.
- **CIGA:** $\lambda_s = [0.2, 0.4, 0.6, 0.8]$, $\alpha = [0.5, 1, 2, 4]$, and $\beta = [0.5, 1, 2, 4]$. $\alpha$ and $\beta$ are for contrastive loss and hinge loss, respectively.

### E.5.2 Hyperparameters of LECI

In GIN-virtualnode cases, we generally apply $\lambda_L \in [10^{-2}, 1]$, $\lambda_E \in [10^{-2}, 10^{-1}]$. In real-world datasets, we adopt $\lambda_{PFSC} \in [10^{-2}, 10^{-1}]$ with a fixed information constraint 0.7 for stable training on GSAT's subgraph discovery basic architecture. Since GOOD-Twitter is too noisy according to the performance, the environment labels on it are far from accurate. Therefore, we extend the search scope as $\lambda_E \in [10^{-3}, 1]$ into the hyperparameter sweeping on GOOD-Twitter.

In all synthetic datasets, to investigate the pure constraints proposed by LECI, we remove the information constraint. For GOOD-Motif basis and GOOD-CMNIST, we apply strong independence constraints, where the searching range becomes $\lambda_L \in [1, 10]$, $\lambda_E \in [10, 100]$, and $\lambda_{PFSC} \in [10^{-1}, 1]$.

The hyperparameter searching scope can be narrowed down using the loss of the two independence components as follows.

Table 10: **Results on the structure and feature shift datasets with the test set hyperparameter selections**

|      | GOOD-Motif | | GOOD-CMNIST | | CFP-Motif | |
| --- | --- | --- | --- | --- | --- | --- |
|      | basis | size | color | covariate | FIIF | PIIF |
| ERM  | 60.93(11.11) | 56.63(7.12) | 26.64(2.37) | 59.33(5.73) | 43.00(7.67) | 62.45(9.21) |
| IRM  | 64.94(4.85) | 54.52(3.27) | 29.63(2.06) | 58.33(5.44) | 68.89(4.99) | 72.89(4.88) |
| VREx | 64.10(8.10) | 57.55(3.99) | 31.94(1.95) | 62.78(2.47) | 44.67(9.73) | 71.00(2.05) |
| Coral | 69.47(1.42) | 60.60(1.40) | 30.79(2.50) | 58.44(9.37) | 51.67(6.60) | 70.44(3.13) |
| DANN | 72.07(1.47) | 59.01(1.55) | 29.36(1.23) | 60.22(3.19) | 52.00(3.14) | 70.00(4.53) |
| ASAP | 45.00(11.66) | 42.23(4.20) | 26.45(2.06) | 64.67(5.63) | 44.34(1.70) | 41.89(5.28) |
| DIR  | 58.49(18.97) | 47.70(10.77) | 26.73(3.25) | 67.33(4.38) | 62.66(9.67) | 57.67(9.27) |
| GSAT | 62.27(8.79) | 57.99(0.91) | 35.02(1.78) | 71.67(7.09) | 55.33(9.69) | 61.22(8.80) |
| CIGA | 67.16(21.59) | 57.10(4.07) | 27.63(3.19) | 60.45(6.38) | 72.33(6.62) | 57.44(9.12) |
| LECI | **86.97(4.48)** | **77.57(4.91)** | **66.75(6.18)** | **88.78(0.69)** | **81.56(2.57)** | **76.67(0.98)** |

**Hyperparameter searching range selection.** There isn't a set of hyperparameters that works perfectly for every dataset. Fortunately, LECI's valid hyperparameter ranges can be easily determined by drawing the loss curve of each component without any test information leaking or extra dataset assumptions. For example, the hyperparameter $\lambda_E$ is valid only when the loss $\mathcal{L}_E$ curve implies a two-stage training pattern. In the first stage, the discriminator should be well-trained, leading to the decrease and convergence of the loss. In the second stage, the adversarial training should take effect; then, the loss will increase. In several simple synthetic datasets, the loss can increase up to the value that indicates a random classification, *e.g.*, this value is $-\log 0.5$ for binary classification. While in real-world datasets, the loss increase will be more gentle.

**Invalid training.** When $\mathcal{L}_E$ or $\mathcal{L}_L$ remain high during the training process, it directly indicates the learning rate or $\lambda_E/\lambda_L$ is too high. In this invalid training case, the model can still produce reasonable results, but LECI does not take effect because this case generally indicates the clear violation of Proposition 3.2. Therefore, different from many other constraints, higher or lower hyperparameters can both lead to invalid training.

**Golden rule.** *Set the hyperparameter searching space around the golden point where the loss $\mathcal{L}_E$ and $\mathcal{L}_L$ fluctuate near their lowest values*, where the lowest values can be measured by setting $\lambda_E$ and $\lambda_L$ to 0. For example, if at the golden point, $\lambda_L = 10^{-1}$ and $\lambda_E = 10^{-2}$, then the searching space is generally $\lambda_L \in [10^{-2}, 10^0], \lambda_L \in [10^{-3}, 10^{-1}]$ except the invalid training processes occurs when we need to narrow the searching space to eliminate them.

This hyperparameter validation process selects the strong constraints for several synthetic datasets, otherwise the loss at the second stage cannot increase to the value that indicates successful adversarial training.

### E.6 Software and hardware

Our implementation is under the architecture of PyTorch [123] and PyG [119]. The deployment environments are Ubuntu 20.04 with 48 Intel(R) Xeon(R) Silver 4214R CPU @ 2.40GHz, 755GB Memory, and graphics cards NVIDIA RTX 2080Ti/NVIDIA RTX A6000.

## F  Supplimentary experiment results

In this section, we provide OOD test set validated sanity check results and interpretability visualizations. For the comparison with GIL [11], we provide it in Appx. B.3.2.

### F.1  A second look at the ability to address structure and feature shifts.

To further exploit the full power of each OOD learning method, we collect the results on the structure and feature sanity check datasets by using **OOD test results to select hyperparameters**. As shown

in Tab. 10, with the leak of OOD test information, several methods improve their performance significantly, yet LECI still markedly outperforms all baselines.

## F.2 Interpretability visualization results

We provide interpretability [38, 98] visualization results on GOOD-Motif and CFP-Motif. As illustrated in Fig. 8, we use thresholds to select the top probability edges as the interpretability subgraph selections. Specifically, on each row, we apply different base subgraphs, including wheel, ladder, tree, path, and small/large Dorogovtsev-Mendes graphs. On each column, we attach the base subgraphs with different motifs, namely, crane, house, and cycle. According to the interpretability results, LECI can select most motifs accurately, which implies its good performance on the causal subgraph selection and further explains the reason behind the good performance. As shown in the figure, the spurious parts of ladders and paths are more challenging to eliminate than other base subgraphs. Fortunately, the latter invariant predictor can distinguish them and make consistent predictions.

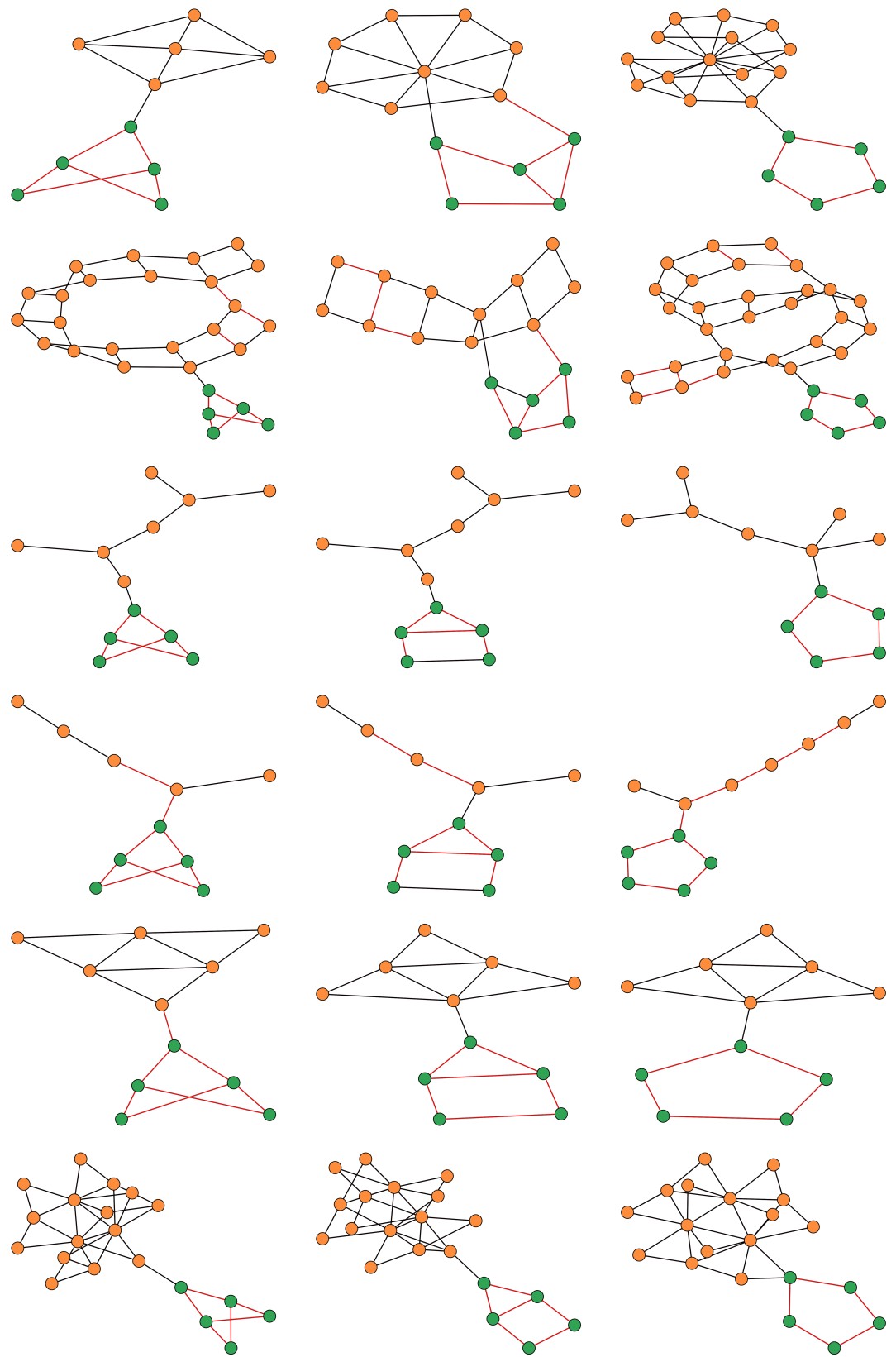

Figure 8: **Interpretability results on GOOD-Motif and CFP-Motif covariate shifts.** Base subgraphs and motifs are denoted as orange and green nodes, respectively. The selected subgraphs are indicated by red edges.

