# OpenReview forum: "Joint Learning of Label and Environment Causal Independence for Graph Out-of-Distribution Generalization"
_NeurIPS.cc/2023/Conference — NeurIPS 2023 poster_

### Official Review · Reviewer_Csim · 2023-07-05

**Soundness:** 2 fair
**Presentation:** 3 good
**Contribution:** 3 good
**Rating:** 5
**Confidence:** 4

**Summary:**

This work addresses the issue of graph out-of-distribution (OOD) generalization by considering the data generation assumption of the presence of a causal subgraph and a spurious subgraph within each graph, and the assumption of label and environment causal independence. The paper proposes an adversarial training approach that leverages accessible environment information in the dataset. By jointly optimizing over these properties, the method aims to discover causal subgraphs for invariant prediction. Experimental evaluations on synthetic and real-world datasets demonstrate the method's superior performance compared to existing approaches.

**Strengths:**

1. The paper is well-written, organized, and effectively conveys its ideas. A notable distinction from prior work is the assumption of access to environmental information, but the authors provide a thorough evaluation of this assumption's fairness compared to existing methods.
2. This paper introduces a novel approach to address the graph OOD problem by integrating environment and label independence, allowing for the comprehensive utilization of environmental information. The proposed method is accompanied by theoretical guarantees, enhancing its credibility.
3. The empirical evaluation is carefully designed, incorporating diverse datasets, relevant baselines, and comprehensive sanity checks. The authors further enhance their study with sensitivity analysis and ablation studies, providing a robust evaluation framework.


**Weaknesses:**

1. The paper introduces assumptions about the data generation process but lacks a formal definition of the OOD task within the paper. It would be beneficial to clarify details such as the training data distribution, test data distribution, and the specific differences between them. Specifically, is the difference solely attributed to the environment variable E, and are new E values observed in the test distribution? Additionally, it would be helpful to address whether the support for $P^\text{tr}(E)$ and $P^\text{te}(E)$ should be the same.
2. Although the paper has theoretical analysis of the proposed approach, it does not directly connect to the OOD generalization error. It is not explicitly stated how the proposed model's performance will be in various OOD settings theoretically.
3. The paper explores three different distribution shift assumptions, but it appears to place more emphasis on the Covariate SCM assumption. In the experimental section, it would be advantageous to clarify the specific SCM that the data satisfy and provide further details regarding how the data satisfy the SCM.


**Questions:**

1. As mentioned in the weakness section, what is the theoretical definition of the graph OOD problem that this paper is trying to tackle?
2. As far as I understand, the theoretical statements claim that if the model is optimized optimally, then the subgraph selector $f_\theta$ should select the optimal causal subgraph in the training graph. However, for the test data with a different distribution, how to guarantee the subgraph selector is still able to find the correct causal subgraph?
3. In line 160, it is said “for any $G_p$ such that $G_s\subset G_p$, $I(G_S;Y) \leq I(G_p; Y)$”. It is unclear whether this statement is an assumption or a proven fact. Since in the line 599, it is said “$I(E; G_C)\geq I(E; G_p)$ is trivial since $G_p$ is a subset of $G_C$”. Why is it trivial? If it is trivial, does it mean “for any $G_p$ such that $G_s\subset G_p, I(G_S;Y) \leq I(G_p; Y) $” is also trivial?
4. In the synthetic dataset section, it seems the dataset follows the Covariant SCM setting. Is it possible to add synthetic experiments for dataset following the FIIF and PIIF settings?
5. Is it possible to know which SCM does each real-world dataset follow?
6. Most real-world datasets show comparable performance across methods, including the ERM baseline. In many cases, the OOD validation performances are better than in-distribution performances. These observations raise questions regarding the difficulty of the OOD task investigated in this section. Can you provide any insights or explanations for these findings?
7. The experimental section primarily focuses on the GOOD benchmark dataset, where the environment information is readily available. Although the paper suggests that obtaining environment labels is feasible by applying the algorithms from GOOD to other datasets, it is worth considering the value and applicability of the environment information in datasets beyond the GOOD benchmark. Clarifying whether the proposed approach can still benefit from the environment information in datasets outside of the benchmark would enhance the understanding of its generalizability.
8. Typo: There are multiple instances of the word "casual" in the paper, which should be corrected to "causal" for accuracy and clarity.


**Limitations:**

1. The components of the method are not particularly novel on their own since it is standard adversarial training. The theoretical guarantee of OOD generalization performance is not fully justified.
2. The accessibility and quality of environment information in general graph datasets are not thoroughly addressed in the paper. Exploring the possibility of incorporating environment information into other existing methods would provide a more comprehensive evaluation and a fairer comparison across different approaches.

---

> ### Author Rebuttal · Authors · 2023-08-09
>
> Thank you for your thoughtful feedback and comments. We have incorporated your suggestions to refine the robustness and clarity of our paper. Below, we provide detailed answers to your questions in order to address your concerns listed in weaknesses.
>
> ## Questions:
>
> 1. We deeply appreciate your request for clarity on this fundamental aspect of our work.
>
> Beyond the preliminary in Sections 2.1 and 2.2, the theoretical definition can be summarized as an interpretation of our SCM assumptions. Essentially, we are addressing the distribution shifts between $P^{tr}(G,Y)$ and $P^{te}(G,Y)$. Our **core assumption**, based on the Independence Causal Mechanism (ICM), is that there exists a component $G_C$ within $G$ such that $P^{tr}(Y|G_C) = P^{te}(Y|G_C)$ (as also discussed in lines 437-446).
>
> The shifts in distribution are exclusively attributed to interventions on $G_S = G - G_C$ that are encapsulated as an environment variable $E$, aligned with the invariant learning literature. The OOD dilemma arises when $P^{tr}(E) \neq P^{te}(E)$, leading to shifts in both $P^{tr}(G_S)$ and $P^{te}(G_S)$, and consequentially, $P^{tr}(G) \neq P^{te}(G)$. It's crucial to note our assumption that both $P^{tr}(G_C)$ and $P^{te}(G_C)$ retain the same support (namely, the **causal subgraph equal support assumption (CSES)**). Explicitly, our target is to resolve OOD scenarios where the supports of $P^{tr}(E)$ and $P^{te}(E)$ differ.
>
> 2. Indeed, your question goes to the core of the subgraph discovery. Let's delve deeper into the mechanics of $f_\theta$ which elucidates a more relaxed form of the aforementioned causal subgraph equal support assumption.
>
> In essence, $f_\theta$ operates as a classifier. When provided with an edge, it classifies the associated ego-graph of the edge into either label 1 (when the central edge is part of $G_C$) or label 0 (otherwise). We denote the distribution of $K$-hop ego-graphs on the edges of $G_C$ as $P(G^{K}\_C)$, and the $K$-hop ego-graph space as $\mathcal{G}^K$. Therefore, we have $f_\theta: \mathcal{G}^K \mapsto [0,1]$, where $\forall G^{K}\_p \in \mathcal{G}^K, f_\theta(G^{K}\_p)=1$ when and only when $P(G^{K}\_C=G^{K}\_p) > 0$, i.e., the distribution of $f_\theta$ is only affected by the support of $P(G^{K}\_C)$. The CSES assumption can be relaxed to that $P^{tr}(G^{K}\_C)$ and $P^{te}(G^{K}\_C)$ have the same support. This ensures the invariance of $f_\theta$ across both training and test distributions. Based on this relaxed assumption, it can be argued that $P(G_C|G)$ remains invariant when $f_\theta$ is employed.
>
> Note that when $K$ is smaller, the assumption is looser. In contrast, when $K→+∞$, this assumption is equivalent to that $P^{tr}(G)$ and $P^{te}(G)$ have the same support, which is stringent. Therefore, $K$ is the "locality rate" for us to loosen a global equal support requirement to a local one.
>
> 3. Clarification: $I(G_S;Y)\le I(G_p;Y)$ is equivalent to $H(Y|G_S) \ge H(Y|G_p)$ where $H$ denotes entropy, since $H(Y)$ is a const. We consider a random graph $G_S$ consist of a set of random variables (edges) $A_S$. $G_S \subseteq G_p$ means $G_p$'s set of random edges $A_p = A_S \cup A_{-}$, where $A_{-}$ is the set of additional edges in $A_p$. Therefore, $H(Y|G_S)=H(Y|A_S)\ge H(Y|A_S,A_{-}) = H(Y|A_p)= H(Y|G_p)$, i.e., $I(G_S;Y)\le I(G_p;Y)$.
>
> 4. Absolutely. We've added **the table in the pdf of the global response.**
>
> 5. You raise an important point. After extensive discussions with domain experts, it's evident that currently, no robust method exists to measure this aspect. It is challenging yet intriguing, and I'm keen to delve deeper into this in my upcoming research.
>
> 6.
> (1) Your observation aligns with findings from DomainBed, where the OOD val results are better than ID val results.
> (2) As shown in the GOOD benchmark's Table 1, the differences in OOD performance on graphs aren't huge, which might relate to insights in [1].
> (3) Another consideration is the expressiveness of GNNs, which might impose constraints on the capabilities of OOD methods.
>
> 7.
> (1) For clarity, both Twitter and DrugOOD are datasets external to the GOOD benchmark, i.e., at least 2 out of 5 real-world datasets lie outside the GOOD benchmark.
> (2) The availability of graph OOD datasets is somewhat limited, and our efforts were made towards encompassing the most widely-recognized graph-level datasets—molecules and languages. These datasets, aside from being utilized in established methods, provide robust empirical support for our claims. If you have specific datasets in mind, we are open to considering their inclusion.
>
> 8. We appreciate your attention to detail. The identified typos have been corrected.
>
> ## Limitations
>
> 1. After refining our theoretical framework and assumptions, we can affirm that the OOD generalization performance is underpinned by the invariant $f_\theta$ and $g_{inv}$.
>
> 2. We understand that our method might not directly correlate with OOD errors. For context, current statistical graph modeling techniques [2, 3] are related to OOD generalization errors, albeit focusing on size shift analysis. Methods inspired by explanations, like DIR and LECI, offer practical solutions capable of addressing more shifts. Nonetheless, bridging these methods to OOD generalization errors presents a substantial challenge, which may not be solved in this single study.
>
> 2. We have experimented with integrating environment information into GIL in B.3.2.
>
> Due to the space limit, please refer to our global response for more limitation discussions.
>
> Sincerely,
>
> Authors
>
> [1] Yang, C. et al. (2022). Graph neural networks are inherently good generalizers: Insights by bridging gnns and mlps. arXiv preprint arXiv:2212.09034.
>
> [2] Bevilacqua, B. et al. (2021, July). Size-invariant graph representations for graph classification extrapolations. In ICML (pp. 837-851). PMLR.
>
> [3] Chu, X. et al. (2023). Wasserstein Barycenter Matching for Graph Size Generalization of Message Passing Neural Networks.

---

> ### Author Response · Authors · 2023-08-20
> **We would like to hear from Reviewer Csim**
>
> Dear Reviewer Csim,
>
> Thanks again for your valuable comments and suggestions, which helps improve our work a lot. Regarding your main concerns on the theoretical definitions and experiments overhead, which we believe is extremely valuable and critical, we have revised our theoretical definitions/assumptions and conducted experiments in our response. Could you please check at your earliest convenience? Thanks!
>
> Since the discussion period is approaching the end, we sincerely hope you can let us know if we have addressed your concerns. Also, we welcome any additional discussions or feedback. Thank you!
>
> Sincerely,
>
> Authors

---

> > ### Comment · Reviewer_Csim · 2023-08-21
> >
> > Dear author,
> >
> > Thank you for your in-depth response. While some of my points have been addressed, I still have doubts about certain theoretical and experimental components of the paper.
> >
> > * In your explanation, you've mentioned that the subgraph selector $f_\theta$ remains consistent in the OOD test graph because it solely determines if a K-hop ego-graph is part of the invariant subgraph. However, even if we assume $P(G_c)\neq P(G_s)$, it doesn’t necessarily mean $G_c$ and $G_s$ do not share similar K-hop subgraph, which might mislead $f_\theta$. For example, if $P(G_c)$ and $P^\text{tr}(G_s)$ do not share similar K-hop subgraph, while $P(G_c)$ and $P^\text{te}(G_s)$ share similar K-hop subgraph, it is possible that $f_\theta$ fail in the OOD setting.
> > * Regarding the experiment, the author did not justify why in many cases the OOD validation performances are better than in-distribution performances, and claimed it could be a property of the dataset and GNNs. These observations raise questions about whether the selected tasks are able to justify the OOD generalization capability of the proposed method and what insight can we gain from the proposed experiment.
> >
> > Thus I will keep my rating as it is for now. However, I remain open to reconsider my rating after discussing with other reviewers.

---

> > > ### Author Response · Authors · 2023-08-21
> > > **Further clarifications**
> > >
> > > Dear Reviewer Csim,
> > >
> > > We appreciate your replies and would like to address your certain concerns and misunderstandings respectfully as follows.
> > >
> > > 1. The core concern is `it doesn't necessarily mean $G_c$ and $G_s$ do not share similar K-hop subgraph, which might mislead $f_\theta$.` To precisely examine whether it is necessary or not, we need to further fix/assume other factors and conditions here: (Note that these conditions are related to the GNN architecture and expressiveness, thus, they won't affect our OOD learning strategy.)
> > > - The number of hop $K$ should be certain value that makes the $K$-hop subgraphs's diameter large enough to cover $G_C$.
> > > - Theoretically, we have an optimal injective $K$-hop subgraph classifier that can distinguish all $K$-hop subgraphs. This implies only the same $K$-hop subgraphs can produce the same result by this classifier, while similar subgraphs cannot.
> > >
> > > **Proposition:** Given the above conditions, there is not a $K$-hop subgraph that shares across $P(G_C)$ and $P^{te}(G_S)$.
> > >
> > > **Proof:** (Contradiction) Given the above conditions, suppose there is a $K$-hop subgraph that shares across $P(G_C)$ and $P^{te}(G_S)$ as described in your example. This indicates that $\exists G^K_p ⊆ P^{tr}(G^K_C)$ such that there is an edge $e_s$ in $P^{te}(G_S)$ whose corresponding $K$-hop subgraph $G^K_{e_s}$(centered with this edge) equals to $G^K_p$, i.e., $G^K_{e_s}=G^K_p$. Since $G^K_p ⊆ P^{tr}(G^K_C)$, the central edge $p$ is an edge on $G_C$, and $G_C\subseteq G^K_p$ that is implied by the first condition above. Since $G^K_{e_s}=G^K_p$, the edge $e_s$ is an edge of $G_C$, and $G_C\subseteq G^K_{e_s}$. Therefore, since our basic assumption indicates an edge can either belong to $G_C$ or $G_S$, and $e_s$ belongs to $G_C$, $e_s$ cannot belong to $G_S$ even though $G_S\in P^{te}(G_S)$, which contradicts the supposition. Therefore, there is no such $K$-hop subgraph that shares across $P(G_C)$ and $P^{te}(G_S)$. **Q.E.D.**
> > >
> > > Therefore, given the above conditions, $f_\theta$ won't be misled by shared $K$-hop subgraphs, since such subgraphs don't exist.
> > >
> > > 2. We sincerely apologize for this confusion and our misunderstanding of the initial review. **The values in Table 2 are not indicating the OOD validation performances and the in-distribution performance. Instead, they both indicate OOD test performances**. The only difference between the columns "ID val" and "OOD val" is that the first one uses in-distribution validation sets **to select model training checkpoints**, while the second one uses OOD validation sets to select. These two sets are only used for validation whose performances are not listed in the table. We hope to clarify this misunderstanding again: *the results shown in the table are the OOD test set performance, which directly indicates the OOD generalization ability of different methods*. Therefore, the insight in the table is that **using OOD validation is generally better than using ID validation in terms of the OOD test performances**. (Note that this message will be emphasized in the caption of the table to avoid confusion.)
> > >
> > > Thank you again for your valuable feedback and suggestions! We hope our explanations and clarifications are satisfying to address these remaining concerns. For the experiment part, if there are further questions, `Reviewer tVF4` may be the appropriate reviewer to discuss with according to the previous review/discussion period.
> > >
> > > Sincerely,
> > >
> > > Authors

---

### Official Review · Reviewer_psug · 2023-07-05

**Soundness:** 2 fair
**Presentation:** 2 fair
**Contribution:** 2 fair
**Rating:** 3
**Confidence:** 4

**Summary:**

This paper proposes a novel adversarial approach called LECI, for OOD generalization. The paper leverages label and environment information to identify the causal and invariant graph, thereby enhancing the accuracy and reliability of causal and invariant subgraph identification. The authors conduct extensive experiments on various real-world datasets under different distribution shift and demonstrate the effectiveness of the proposed approach.

**Strengths:**

(1) The paper proposes an effective learning strategy to incorporate label and environment causal independence for tackling the graph OOD problem.

(2) Detailed theoretical guarantees for casual subgraph discovery are provided.

(3) Extensive experiments on synthetic and real-world datasets, which provide insightful empirical findings.



**Weaknesses:**

1. Limited novelty. Since environment inference and environment exploitation are often combined, and previous methods have used environment-based approaches specifically for graphs[1-2], it is puzzling why the author propose that previous methods did not explicitly mention graph-specific designs in line 26. Moreover, if representations are obtained by both structural and feature information using message passing, it begs the question of whether invariant learning on such representations has already addressed by previous works. The novelty of the article requires more presentation.

2. Tables 1 and 2 do not include a comparison with GIL[2] as mentioned in line 266. Furthermore, based on my limited knowledge, GIL also involves capturing and optimizing invariant subgraphs, utilizing invariant learning for optimization. It is recommended to provide an explanation regarding the explainability of subgraphs between LECI and GIL. It appears that the paper requires additional elaboration on its novelty.

3.Does the presence of three relatively independent adversarial training criteria in the main model pose significant challenges to the model training, such as difficulties in convergence similar to DANN? From Figure 4 on the left, a similar trend can be observed. It seems that this model requires dedicated hyper-parameters such as learning rate and number of epochs in order to achieve better results.

[1] Chen, Yongqiang, et al. "Learning causally invariant representations for out-of-distribution generalization on graphs."  NIPS, 2022.

[2] Li, Haoyang, et al. "Learning invariant graph representations for out-of-distribution generalization." NIPS, 2022.



**Questions:**

see Weaknesses

---

> ### Author Rebuttal · Authors · 2023-08-08
>
> We sincerely appreciate the effort and time you have dedicated to reviewing our paper. We understand that some aspects may have led to certain misconceptions, and it is our foremost intention to address and clarify these points. We hope this further elucidation will assist in reevaluating our work.
>
> ## Weaknesses:
>
> > Limited novelty. Since environment inference and environment exploitation are often combined, and previous methods have used environment-based approaches specifically for graphs[1-2], it is puzzling why the author propose that previous methods did not explicitly mention graph-specific designs in line 26.
>
> **Response:** Thank you for pointing out the concerns related to the novelty and positioning of our work in relation to existing literature.
>
> 1. While we acknowledge the contributions of [1], it fundamentally operates as an environment-agnostic method. One can observe this from Figure 1(b) in [1], even though they did delve into analyses that involve environments.
>
> 2. The ambiguity in line 26 is duly noted, and we appreciate your suggestion. To elaborate:
>    - Our stance in line 26 is elucidated in lines 76-88.
>    - For a comparative understanding with environment-based methods including [2], we direct your attention to section B.3.1 in the appendix.
>    - For a comparison with other methodologies, section C.2 in the appendix sheds light.
>
> Our intention was to underline unexplored avenues in this domain, and the references and justifications we've provided aim to demarcate our contributions distinctly. We remain open to further discussions and clarifications on this matter.
>
> > Moreover, if representations are obtained by both structural and feature information using message passing, it begs the question of whether invariant learning on such representations has already addressed by previous works. The novelty of the article requires more presentation.
>
> **Response:** Thank you for highlighting the need for clarity on the novelty of our approach, especially in the context of invariant learning on combined structural and feature representations.
>
> To specifically address this concern:
> - We delve deep into this topic in section B.3.2 of the appendix. Therein, we performed experiments that hold all other variables constant, allowing for a direct comparison between graph-specific (our method, LECI) and non-graph-specific (like GIL/IGA) environment exploitation strategies.
> - The findings conclusively indicate that merely applying invariant learning on the combined representation (of structure and feature) falls short in addressing the challenges, and in fact, produces suboptimal outcomes in contrast to our graph-specific exploitation approach.
>
> We believe that this rigorous experimental comparison further underscores the novelty and significance of our proposed method.
>
> > Tables 1 and 2 do not include a comparison with GIL[2] as mentioned in line 266. Furthermore, based on my limited knowledge, GIL also involves capturing and optimizing invariant subgraphs, utilizing invariant learning for optimization. It is recommended to provide an explanation regarding the explainability of subgraphs between LECI and GIL. It appears that the paper requires additional elaboration on its novelty.
>
> **Response:** Thank you for pointing it out, and we appreciate the opportunity to clarify:
>
> 1. Firstly, our discussions and justification for the novelty, particularly in the context of GIL, can be found in section B.3.1 of the appendix. We believe this section provides a thorough insight into our approach and its distinctiveness.
>
> 2. As for the direct comparison with GIL as noted in line 266, due to space constraints in the initial submission and the added complexity of capturing environment exploitation comparisons, the results were shifted to B.3.2 in the appendix. Please be assured that, upon acceptance, we will reintegrate these comparisons into the main content of the paper to provide a clearer and more comprehensive perspective.
>
> We strive for clarity and completeness in our presentation and appreciate your feedback which aids us in achieving that.
>
> > Does the presence of three relatively independent adversarial training criteria in the main model pose significant challenges to the model training, such as difficulties in convergence similar to DANN? From Figure 4 on the left, a similar trend can be observed. It seems that this model requires dedicated hyper-parameters such as learning rate and number of epochs in order to achieve better results.
>
> **Response:** We acknowledge your concern regarding the potential challenges posed by the inclusion of adversarial training criteria in the primary model. Indeed, the complexities associated with such a setup could be reminiscent of those seen in DANN, as evidenced by the trend in Figure 4 on the left.
>
> However, it's crucial to emphasize that while these challenges are present, they don't substantially hamper the validity of our claims or the conclusions we draw from our experiments. For a more comprehensive understanding of these challenges and our response to them, we refer you to our global response section (under common concerns & limitations).
>
> Furthermore, detailed specifications of our chosen hyper-parameters can be found in section E.5.1 of the appendix. A noteworthy observation, as shown in Table 10, is that **even with hyper-parameter optimization using the test set, other existing methods still do not manage to match LECI's performance**. This underscores the notion that our approach's merits aren't product of hyper-parameter tuning but instead arise from its inherent design and functionality.
>
> Sincerely,
>
> Authors
>
> ## References
>
> [1] Chen, Yongqiang, et al. "Learning causally invariant representations for out-of-distribution generalization on graphs." NIPS, 2022.
>
> [2] Li, Haoyang, et al. "Learning invariant graph representations for out-of-distribution generalization." NIPS, 2022.

---

> > ### Comment · Reviewer_psug · 2023-08-17
> > **Response to authors**
> >
> > Thank you for the authors' response. The authors have addressed some of my concerns; however, there is still room for improvement regarding the novelty aspect. The idea of using the environment to enhance generalization has already been explored in various previous works.

---

> > > ### Author Response · Authors · 2023-08-17
> > > **Further response and discussions on novelty**
> > >
> > > Dear Reviewer psug,
> > >
> > > Thank you for your reply! We're truly appreciative of the time and effort you've invested in reviewing our work.
> > > As mentioned, your remaining concern is in the novelty regarding previous works using environments.
> > > To elucidate our novelty, we have dedicated to emphasizing distinctions to previous studies.
> > > We focus on eliminating your concern and clearly discuss our novelty from the following perspectives.
> > >
> > >
> > > - **Position**: Our work, LECI, stands out as **the first graph-specific environment exploitation** technique. (Justification is as Table 4 in the Appendix.) We would like to emphasize that the uniqueness resides in both 'graph-specific' and 'environment exploitation', and our idea is above simply using environments. Using environment labels in graphs reveals unique challenges and opportunities specific to graph tasks that require innovative designs, theoretical results, and empirical validations. Though many general OOD methods use environment as a common characteristic, **our work of graph-specific design can be substantially distinguished from existing works**.
> > >
> > > - **Challenges & Theory**: LECI explores the subgraph discovery challenges and examines the boundaries and limitations of prior assumptions. Accordingly, we propose **sufficient and necessary theoretical guarantees** that address these challenges - a clear testimony to the novelty of our approach.
> > > - **Experimental Studies**: Setting a precedent, LECI is the **first method to successfully pass the structure shift sanity check**. Its significant and robust performance is observed through diverse experiment designs, including synthetic datasets, real-world datasets, ablation study, sensitivity analysis, and environment exploitation comparisons (graph-specific v.s. non-graph-specific in Sec. B.3.2).
> > > - **Community Impact**: Contrary to a prevalent belief in the community, as referenced in [1], we have showcased that environment information, far from being too noisy, can be a potent tool in graph tasks, which is evidenced more powerful than previous inductive biases (or assumptions). The caveat that previous general environment exploitation methods cannot work is the lack of graph-specific design for solving specific challenges occurring in graph tasks.
> > >
> > > In essence, the novelty of our work can be distilled as below:
> > > 1. A unique positionality in the domain.
> > > 2. Novel theoretical insights and problem-solving.
> > > 3. Comprehensive, varied, and robust experimental validations.
> > > 4. Challenging and reshaping established community perspectives.
> > >
> > > We have made much efforts to justify each of the above points in detail in our paper, and we reiterate them in this response and the initial/global rebuttal for clarity.
> > >
> > > Your feedback is valuable, and we aim to incorporate it in refining our related work section to further underscore our contributions in novelty. We welcome more questions and would be grateful if they come with contextual justifications, allowing us to address them effectively.
> > >
> > > Sincerely,
> > >
> > > Authors
> > >
> > > [1] Yang, N., Zeng, K., Wu, Q., Jia, X., & Yan, J. (2022). Learning substructure invariance for out-of-distribution molecular representations. Advances in Neural Information Processing Systems, 35, 12964-12978.

---

> > > ### Author Response · Authors · 2023-08-20
> > > **Author's follow up to reviewer psug**
> > >
> > > Dear Reviewer psug,
> > >
> > > Thanks again for your valuable comments and suggestions. Regarding your remaining concerns on novelty, we have made efforts to address these critical points. Could you please check the responses at your earliest convenience? Thanks!
> > >
> > > Since the discussion period is approaching the end, we sincerely hope you can let us know if we have addressed your concerns and reevaluate if we do. Also, we welcome any additional discussions or feedback with contextual justifications. Thank you!
> > >
> > > Sincerely,
> > >
> > > Authors

---

### Official Review · Reviewer_YxXK · 2023-07-06

**Soundness:** 3 good
**Presentation:** 3 good
**Contribution:** 3 good
**Rating:** 5
**Confidence:** 4

**Summary:**

The paper proposed a novel learning strategy to incorporate label and environment causal independence (LECI) for tackling the graph OOD problem. Enforcing such independence properties can help exploit environment information and alleviate the challenging issue of graph topology shifts. With good empirical results on several datasets and related theoretical analyses, the paper justifies the effectiveness of the proposed LECI.

**Strengths:**

1.	The paper proposed an effective method for tackling the graph OOD problem. The author proposes to solve the problem by introducing label and environment causal independence, which is a good idea.
2.	The logic is clear and solid, including the well-defined research problem and challenges, the proposed solutions, and theoretical analysis.
3.	The gained empirical improvement is quite significant with the proposed LECI framework. Several ablation studies are shown.


**Weaknesses:**

1.	I am puzzled for "graph-specific and "distribution shift on graph topology". Compared with other graph-specific OOD baselines, why LECI can solve the problem, as the author mentioned on line 19 and line 20.
2.	It would be better to perform analysis on the complexity analysis, and some details are not explained so clearly of the two discriminators.
3. Due to the adversarial mechanism, the proposed method cannot scale to large-scale datasets and also it is very hard to train.


**Questions:**

In addition to the questions given above, I also want to ask some additional questions.

4. The question is related to the adversarial mechanism. Take EA as an example. For both equations 6 and 7, the adversarial principles are easy to understand. However, From the mathmatical perspective, I am lost. For equation 6, the inner max can be naturally converted to min L_E, as that in Equation 7. In this way, with the same inner function, the outer operation in Equation 6 is min while that in Equation 7 is max. Is this a contradiction?

5. From figure 4, the curves of LECI drop first and then increase, which is marked by the effect of the two independence constraints. Why do the two independence constraints take effect too late?

**Limitations:**

The authors provide detailed analysis on limitations in the supplementary materials.

---

> ### Author Rebuttal · Authors · 2023-08-08
>
> We appreciate your valuable comments and feedbacks! We have carefully considered each one and have endeavored to address your concerns. We believe that our responses will provide the necessary clarifications, and we look forward to your feedbacks.
>
> ## Weaknesses:
>
> > I am puzzled for "graph-specific and "distribution shift on graph topology". Compared with other graph-specific OOD baselines, why LECI can solve the problem, as the author mentioned on line 19 and line 20.
>
> **Response:** We recognize the ambiguity and appreciate your patience. We are committed to enhancing clarity in the paper by incorporating pertinent references. Here’s a breakdown of our clarification:
>
> 1. In relation to environment-based methods: Please consult section B.3.1 in the appendix. Here, we elucidate that LECI uniquely harnesses environment information in graph contexts in a manner that other methods don't. B.3.2 further demonstrates the result differences.
>
> 2. Regarding environment-agnostic approaches:
>     * First, refer to section C.2 and Table 8 in the appendix. This demonstrates that the assumptions guiding other techniques are more susceptible to being compromised compared to using environment information.
>     * Second, regarding the "distribution shift on graph topology", please see Table 7 and section C.2 in the appendix. We highlight therein that other strategies falter in addressing dual subgraph discovery challenges, a feat LECI manages to achieve.
>
> 3. For a deeper dive into these subgraph discovery challenges, C.3 in the appendix provides comprehensive insights.
>
> > It would be better to perform analysis on the complexity analysis, and some details are not explained so clearly of the two discriminators.
>
>
> **Response:** We genuinely appreciate this constructive feedback. To address this point and similar concerns, we've elaborated in our global rebuttal (under the common concerns section).
>
> We kindly seek further specificity regarding which details you'd like elaborated about our discriminators, especially beyond our explanations provided in Eq. (7) and Eq. (8). We've made our code available as supplementary material, which should provide comprehensive insight into the detailed workings. However, if you believe there are critical aspects to be incorporated into the paper itself, we stand ready to make those modifications.
>
> > Due to the adversarial mechanism, the proposed method cannot scale to large-scale datasets and also it is very hard to train.
>
> **Response:** Indeed, your observation regarding the limitations posed by the standard adversarial mechanism is astute. We have explicitly discussed this point in section C.4 of our appendix. Nevertheless, it's crucial to emphasize that this specific limitation does not detract from the primary claims and conclusions of our study, as underscored in our global response.
>
> ## Questions:
>
> > The question is related to the adversarial mechanism. Take EA as an example. For both equations 6 and 7, the adversarial principles are easy to understand. However, From the mathmatical perspective, I am lost. For equation 6, the inner max can be naturally converted to min L_E, as that in Equation 7. In this way, with the same inner function, the outer operation in Equation 6 is min while that in Equation 7 is max. Is this a contradiction?
>
> **Response:** Thank you for your question. To clarify, the equations operate as follows:
>
> For Eq. (6), the operative function is:
> $$\theta^*_\text{E} = argmin_{\theta} \left\\{\max_{\phi_E} \mathbb{E} \left[ \log{P_{\phi_E}(E|\hat{G}_C)} \right]\right\\}$$
>
> However, if we introduce a negation within the argmin, it translates to argmax:
>
> $$argmin\_{\theta} \left\\{\max_{\phi_E} \mathbb{E} \left[ \log{P_{\phi_\text{E}}(E|\hat{G}\_C)} \right]\right\\} = argmax_{\theta} \left\\{ - \max_{\phi_E} \mathbb{E} \left[ \log{P_{\phi_\text{E}}(E|\hat{G}_C)} \right]\right\\}$$
>
> Upon transposing the negative sign and the max operation:
>
> $$ argmax_{\theta} \left\\{ - \max_{\phi_E} \mathbb{E} \left[ \log{P_{\phi_\text{E}}(E|\hat{G}\_C)} \right]\right\\} = argmax_{\theta} \left\\{\min_{\phi_E} - \mathbb{E} \left[ \log{P_{\phi_\text{E}}(E|\hat{G}_C)} \right]\right\\} $$
>
> Your assertion "For equation 6, the inner max can be naturally converted to min L_E" is partially accurate. Yes, in the context of the optimal $\phi_E$, it holds that
>
> $$argmax_{\phi_E} \mathbb{E} \left[ \log{P_{\phi_\text{E}}(E|\hat{G}\_C)} \right] = argmin_{\phi_E} L_E. $$
>
> However, the overall optimality of their values contrast, hence
>
> $$ max_{\phi_E} \mathbb{E} \left[ \log{P_{\phi_\text{E}}(E|\hat{G}\_C)} \right] = - min_{\phi_E} L_E. $$
>
> This inherent "minus" justifies the outer argmax in Equation 7.
>
> > From figure 4, the curves of LECI drop first and then increase, which is marked by the effect of the two independence constraints. Why do the two independence constraints take effect too late?
>
> **Response:** We've addressed this in our global rebuttal under the common concerns section, and we thank you for highlighting it.
>
> Sincerely,
>
> Authors

---

> ### Author Response · Authors · 2023-08-20
> **We would like to hear from Reviewer YxXK**
>
> Dear Reviewer YxXK,
>
> Thanks again for your valuable comments and suggestions, which helps improve our work a lot. Regarding your constructive concerns and suggestions on the method and complexity analysis overhead, we have made revisions and clarifications in our response. Could you please check at your earliest convenience? Thanks!
>
> Since the discussion period is approaching the end, we sincerely hope you can let us know if we have addressed your concerns. Also, we welcome any additional discussions or feedback. Thank you!
>
> Sincerely,
>
> Authors

---

### Official Review · Reviewer_6JF6 · 2023-07-06

**Soundness:** 3 good
**Presentation:** 3 good
**Contribution:** 3 good
**Rating:** 6
**Confidence:** 3

**Summary:**

This paper proposes a new graph learning model to incorporate label and environment causal independence (LECI) for solving the graph OOD generalization problem. The authors first introduce two causal independence properties to distinguish causal and spurious subgraphs. Enforcing environment causal independence properties and utilizing the environment label information, LECI is able to identify causal subgraphs for invariant predictions. The authors further leverage an adversarial learning strategy to learn the label and environment casual independence. In contrast to the previous graph OOD methods which commonly assume non-existence of the environment information, LECI instead fully utilizes environment information and introduces a graph-specific environmental exploitation algorithm. Extensive experiments on GOOD and DrugOOD benchmark datasets demonstrate the superiority of LECI among other baseline methods.


**Strengths:**

1. The illustration of graph OOD generalization from a causal perspective is clear and easy to understand, and the derived two causal independence properties could be used to distinguish between causal subgraphs and spurious subgraphs for graph OOD future works.
2. The adversarial learning implementation of LECI is technically sound, and the derivation of the adversarial objective is rigorous.
3. The theoretical results are convincing.
4. Extensive experiments on GOOD and DrugOOD benchmark datasets demonstrate that LECI significantly outperforms baselines, and the authors also conducted comprehensive experiments for sensitivity analysis and ablation studies.


**Weaknesses:**

1. LECI assumes the environment information is known, which is a tight requirement. The authors argue that the environment labels can be accessed by applying simple groupings or deterministic algorithms, but in this case, the obtained labels are not accurate. From my perspective, these labels can be served as auxiliary information or as a proxy for ground-truth environment labels, but directly regarding them as environment labels may be not appropriate. However, I do believe LECI can be combined with the existing environment inference methods. I wonder if the authors have made some attempts.
2. The authors did not discuss the computational complexity of LECI in the paper.


**Questions:**

1. Questions in Weaknesses part.
2. As environment labels may only be observed partially in some real-world scenarios, I wonder if LECI could utilize the observed environment labels to infer the rest of the labels using the adversarial implementation.


**Limitations:**

The authors adequately addressed the limitations and future work in the Conclusion section.

---

> ### Author Rebuttal · Authors · 2023-08-08
>
> We appreciate your helpful feedbacks!
>
> ## Weaknesses:
>
> > LECI assumes the environment information is known, which is a tight requirement. The authors argue that the environment labels can be accessed by applying simple groupings or deterministic algorithms, but in this case, the obtained labels are not accurate. From my perspective, these labels can be served as auxiliary information or as a proxy for ground-truth environment labels, but directly regarding them as environment labels may be not appropriate. However, I do believe LECI can be combined with the existing environment inference methods. I wonder if the authors have made some attempts.
>
> **Response:** Following your suggestion, we evaluated LECI alongside a cluster-style environment inference (EI) approach that employs k-means on the representations of $G_S$, treating their cluster labels as environment labels. We gathered results across two real-world datasets and one synthetic dataset:
>
> |                | HIV-scaffold | Twitter | Motif |
> | -------------- | ------------ | ------- | ----- |
> | LECI + cluster | 71.38        | 57.17   | 72.67 |
>
>
> It's crucial to emphasize that without a comprehensive theoretical design to ensure subgraph discovery, this approach doesn't surpass our current LECI implementation that uses environment labels. This can be attributed to challenges like selecting appropriate EI methods and determining the number of environments, etc. We share these results to validate that LECI is indeed compatible with EI methods. Nonetheless, a deep exploration of EI methods is beyond the scope of this study. We acknowledge this and will reference it as a limitation in Section C.4 of the appendix.
>
> > The authors did not discuss the computational complexity of LECI in the paper.
>
> **Response:** Thank you for your suggestion. We include the response in our global rebuttal (the common concerns section).
>
> ## Questions:
>
> > As environment labels may only be observed partially in some real-world scenarios, I wonder if LECI could utilize the observed environment labels to infer the rest of the labels using the adversarial implementation.
>
> **Response:** Theoretically, LECI is indeed capable of utilizing the environment discriminator to predict additional environment labels from those that are observed. However, in real-world situations, we often encounter class-incremental tasks in environment inference, where the test environment labels might differ substantially from the observed ones. This creates a challenge that our environment discriminator is not designed to address. To illustrate, consider a scenario where during training, the model learns to differentiate environments $e_1, e_2, e_3$, yet in the testing phase, it encounters environments $e_3, e_4, e_5$. Here, $e_4$ and $e_5$ are unknown quantities, and their prediction would require specialized class-incremental techniques.
>
> Sincerely,
>
> Authors

---

> > ### Comment · Reviewer_6JF6 · 2023-08-17
> > **Re: Rebuttal by Authors**
> >
> > Thank you for the rebuttal. The authors addressed some of my concerns, as they evaluated LECI together with a K-means cluster-style environment inference (EI) approach on several datasets. However, I would like to keep my score as is, since the results in the rebuttal are preliminary, and the authors also acknowledged that a deep exploration of EI methods is needed for future study. Moreover, thank the authors for answering my question. I agree that class-incremental tasks are often encountered and specialized techniques should be designed.

---

> > > ### Author Response · Authors · 2023-08-17
> > >
> > > Dear Reviewer 6JF6,
> > >
> > > Thank you for your kind reply and recognition of our paper! We completely agree that developing environment inference (EI) methods to incorporate with LECI is an interesting point. Since we focus on the establishment of the `first graph-specific environment exploitation method` in this paper, this initial work might not cover further EI tasks. It is essential to solve subgraph discovery challenges before more EI methods can be effectively explored. We will add possible EI methods in our Discussion section to benefit future explorations.
> > >
> > > We sincerely thank you for your time and your acknowledgment of our contributions! Hope we have addressed your concerns through practical efforts and shown the significance of our work. And we look forward to any further discussions, thanks!
> > >
> > > Sincerely,
> > >
> > > Authors

---

### Official Review · Reviewer_tVF4 · 2023-07-06

**Soundness:** 3 good
**Presentation:** 3 good
**Contribution:** 3 good
**Rating:** 6
**Confidence:** 4

**Summary:**

This papers considers the graph OOD generalization problem with given environment information. It proposes a novel invariant learning method based on adversarial loss which is derived from independence conditions of the causal graph. Experimental results show the proposed method is very effective.


**Strengths:**

- Compared with previous work, this work proposes a more graph-specific environmental exploitation algorithm, which includes a subgraph selector.

- The proposed method comes with theoretical results to show that the derived optimal paramteres can lead to learning causal subgraph only for invariant prediction.

- The proposed method outperforms existing methods on various benchmarks by a large margin. Ablation studies show the proposed method is benefiting from each component.


**Weaknesses:**

- In fact, the proposed method can also be not that graph specific. We only need to replace the subgraph selector f_{\theta} with a feature selector, then we can apply the method to invariant rationalization [1]. So, the claim may not be 100% true.

- The theorems only provide equivalence between optimal paramters and learning causal subgraphs, but they did not mention what can be guaranteed if we cannot achieve optimal parameters (e.g., due to hidden variables).

[1] https://arxiv.org/abs/2003.09772


**Questions:**

- Since Y is not necessarily independent of G_s in FIIF and PIIF, what is the motivation to even use a soft version of the independence property?

- For model selection, we know most invariant learning methods do not work with in-domain validation set [1] for image datasets. I am curious about what is the overall improvement when using OOD validation set over in-domain validation set in the graph OOD problem.

[1] Gulrajani, Ishaan, and David Lopez-Paz. "In search of lost domain generalization." arXiv preprint arXiv:2007.01434 (2020).


**Limitations:**

- This work only considers causal features and spurious features as subgraphs. But there can other types of high-level graph features that can be causal features.

- This work relies on invariant learning, which itself can be limited. For example, when the invariant features are not highly correlated with label, then the prediction accuracy can be quite low.

---

> ### Author Rebuttal · Authors · 2023-08-08
>
> Thank you for your valuable time and insightful comments!
>
> ## Weaknesses:
>
> > In fact, the proposed method ... the claim may not be 100% true.
>
> **Response:** As highlighted on line 564 of the appendix, the mapping from $(X,A)_C \mapsto X_C$ is non-unique. While it's conceivable to adapt a feature-only method to a combined feature+structure method, the challenges encountered differ significantly. In our approach, we disentangle $A$ into $A_C$ and $A_S$, but refrain from dissecting $X$. In comparison, [1] postulates that $X$ can be disentangled, an assumption that may not hold for graph node embeddings. Thus, our methodology indeed demonstrates graph-specific attributes when juxtaposed with other graph OOD techniques. Nevertheless, in light of your feedback, we are committed to providing further elucidation and will moderate our claim for precision. We trust you appreciate that this assertion helps to distinctly position our work within the graph OOD discourse.
>
>
> > The theorems only provide ... what can be guaranteed if we cannot achieve optimal parameters (e.g., due to hidden variables).
>
> **Response:** Should the optimal parameters remain unattainable, our theorems indicate an associated increase in loss. We recognize the complexities introduced by bi-level optimization, analogous to those observed in GANs. There exist subsequent studies, such as WGAN and consensus optimization methods, dedicated to addressing these optimization challenges. While this particular aspect isn't central to our paper's core message and claims, we acknowledge its significance. As a gesture towards its importance, we have identified it as a noteworthy limitation warranting future exploration and have documented this perspective in line 692 of the appendix.
>
>
> We hope our answers address your concerns about our paper.
>
> ## Questions:
>
> > Since Y is not necessarily independent of G_s in FIIF and PIIF, what is the motivation to even use a soft version of the independence property?
>
> **Response:** The soft independence property encompasses the strict version of the independence property as a specific instance when $I(G_S;Y)=0$. As elucidated in Theorem 3.8, the soft independence property buttresses our learning approach. Transitioning from the strict "independence property" to the softer version serves a dual purpose: it not only structures our paper coherently but also progressively guides the readers from simpler covariate SCMs (with strict independence) to the more intricate FIIF/PIIF SCMs (featuring soft independence).
>
>
> > For model selection, ... what is the overall improvement when using OOD validation set over in-domain validation set in the graph OOD problem.
>
> **Response:** For model parameter selection, we've incorporated a table delineating the performance difference from ID to OOD model selections of leading-edge methods across real-world datasets like GOODHIV, GOODSST2, and Twitter. Should it be deemed essential, we stand ready to complete and append a comprehensive table to our appendix.
>
> | | SST2 | Twitter | HIV-scaffold | HIV-size |
> | -- | -- | ---- | ---- | --- |
> | GSAT | 0.08 | 0 | 0.48 | 2.29 |
> | CIGA | 0.57 | 0 | 1.87 | 4.7 |
> | LECI | 1.48 | -0.2 | 1.04 | 1.18 |
>
> In the case of model checkpoint selections in a single run, the performance difference can be found in Table 2 in our paper.
>
> ## Limitations:
>
> > This work ... causal features.
>
> **Response:** Indeed, the scope of our work is primarily centered around causal and spurious features modeled as subgraphs because of the graph modeling we used. In fact, the broader landscape of graph modeling, be it statistical or inspired by explainability, is vast, but statistical modeling methods currently can only tackle size shifts [2,3]. While we have dedicated a segment to feature shifts in Sec. 3.7, it's acknowledged that there are more intricate, high-level graph features that could bear causal significance.
>
>
> > This work relies ... can be quite low.
>
> **Response:** Your observation delves into the profound intricacies of causality in real-world scenarios. Invariant learning, as we've outlined in line 437 of the appendix, draws its inspiration from the annals of causality literature. The scenario you describe, where invariant features may not be observed or cataloged, is intimately tied to data collection methodologies. When our data sources are not comprehensive, the consequent statistical rules we deduce might be flawed. Instances where only spurious features are in focus exemplify an anti-causal assumption, which, while encapsulated by invariant learning, is not the focus of our paper. Your example underscores the nuances of causality assumptions and their applicability. We've delved deeper into this topic in section C.2 of the appendix. It's our assertion that while individual generalization challenges can be tackled piecemeal, a holistic, universal solution, as posited by [4], might be elusive.
>
> Both points you've raised are indeed pertinent challenges, meriting deeper exploration in subsequent works. We are committed to weaving these insights into our limitation discussion, specifically in section C.4 of the appendix.
>
> Sincerely,
>
> Authors
>
> ## References
>
> (They will also be added in our paper.)
>
> [1] Chang, S., Zhang, Y., Yu, M., & Jaakkola, T. (2020, November). Invariant rationalization. In International Conference on Machine Learning (pp. 1448-1458). PMLR.
>
> [2] Bevilacqua, B., Zhou, Y., & Ribeiro, B. (2021, July). Size-invariant graph representations for graph classification extrapolations. In International Conference on Machine Learning (pp. 837-851). PMLR.
>
> [3] Chu, X., Jin, Y., Wang, X., Zhang, S., Wang, Y., Zhu, W., & Mei, H. (2023). Wasserstein Barycenter Matching for Graph Size Generalization of Message Passing Neural Networks.
>
> [4] Kaur, J. N., Kiciman, E., & Sharma, A. (2022). Modeling the data-generating process is necessary for out-of-distribution generalization. arXiv preprint arXiv:2206.07837.

---

> > ### Comment · Reviewer_tVF4 · 2023-08-18
> > **thanks for the response**
> >
> > Thanks for the response, authors addressed most of my concerns.
> >
> > > Response: For model parameter selection, we've incorporated a table delineating the performance difference from ID to OOD model selections of leading-edge methods across real-world datasets like GOODHIV, GOODSST2, and Twitter. Should it be deemed essential, we stand ready to complete and append a comprehensive table to our appendix.
> > - For the table shown by the authors, it looks to me the Twitter dataset is an outlier, any insight for that?

---

> > > ### Author Response · Authors · 2023-08-19
> > >
> > > Dear Reviewer tVF4,
> > >
> > > Thank you for replying and we are glad to know that the majority of your concerns have been addressed. Your observation about the Twitter dataset being an outlier is astute, and we appreciate the chance to delve deeper into this aspect.
> > >
> > > To explore the reason, we first list the ID test performances of GOODHIV, GOODSST2, and Twitter:
> > >
> > > | Dataset | GOODSST2 | Twitter | GOODHIV-scaffold | GOODHIV-size |
> > > | ------- | -------- | ------- | ---------------- | ------------ |
> > > | ID test | 91.62    | 68.11   | 80.87            | 82.59        |
> > >
> > > From the table, it's evident that the Twitter dataset's ID test performance is notably lower, suggesting that the Twitter dataset might be inherently noisier.
> > >
> > > Considering the VC-dimension-based transfer learning generalization bound [1], the performance on the target (OOD) distribution is affected by both source (ID) performance and the magnitude of shifts between source (ID) and target (OOD) distributions. **In Twitter's case, given the relatively large ID error, the model's OOD performance can be impeded by its ID performance. Hence, the use of ID validation in such scenarios becomes beneficial.**
> > >
> > > Note that an OOD validation distribution isn't a direct reflection of the OOD test distribution. Rather, it leans in-between the ID and OOD test distributions.
> > >
> > > Therefore, the values "0" and "-0.2" we referenced in our rebuttal can be caused by a combination of factors: the disparity between the OOD validation and test distributions, the limited ID performance, and inherent randomness.
> > >
> > > When the ID error is low, using OOD validation is advantageous. However, drawing from this discussion, when ID error is high, selecting model hyperparameters using both ID and OOD validation sets can be another reasonable choice.
> > >
> > > Your probing question has illuminated a valuable avenue for discussion. We'll ensure these insights find their way into our paper, specifically in the experimental discussions following Sec. 4.4.
> > >
> > > Sincerely,
> > >
> > > Authors
> > >
> > > [1] Ben-David, S., Blitzer, J., Crammer, K., Kulesza, A., Pereira, F., & Vaughan, J. W. (2010). A theory of learning from different domains. Machine learning, 79, 151-175.

---

### Author Rebuttal · Authors · 2023-08-08

# Rebuttal of LECI

## The strengths of this paper

We would like to extend our gratitude to all the reviewers for their invaluable time and insightful feedback. It is encouraging to observe that our contributions and strengths resonated positively with you:

- Novelty (Reviewer tVF4, YxXK, Csim)
- Presentation & Organizaion (Reviewer 6JF6, Csim)
- Motivations & Challenges & Problem formulation (Reviewer 6JF6, YxXK, Csim)
- Graph-specific exploitation method (Reviewer tVF4, 6JF6, YxXK, Csim)
- Theoretical results (Reviewer tVF4, 6JF6, YxXK, psug, Csim)
- Significant and diverse experimental results (Reviewer tVF4, 6JF6, YxXK, psug, Csim)
- Assumption's fairness comparisons (Reviewer Csim)

## The main message this paper can convey to the graph OOD community:

Utilizing a **graph-specific environment exploitation** method with **simply annotated environment partitions** can outperform all contemporary methods rooted in **refined human-made assumptions** (inductive bias), which underscores the need to reevaluate the practical implications of existing assumptions.

## The position of this paper

- The innovative graph-specific environment exploitation method (line 26)
- The first graph OOD method that passes structural shift sanity checks (line 294)

We recognize, as highlighted by Reviewer psug, that the absence of clear reading guidance in line 26 might have given rise to certain misinterpretations and concerns. To preclude any further misunderstandings, we present a comprehensive claim support list for any reviewers who might share these concerns or have related queries:
1. Extentions of the claim in line 26: line 76-88
2. Compared with environment-based methods (detailed claim justifications): Sec. B.3.1 in the appendix
3. Compared with environment-agnostic methods : Sec. C.2 and B.2 in the appendix.
4. Subgraph discovery challenge solution comparisons: Table 7 in the appendix.
5. Result comparisons between non-graph-specific and graph-specific environment exploitation methods under a fair/variable controlled setting: Sec. B.3.2 in the appendix.

This guidance will be integrated into the introduction of the paper.

## Common concerns

### Computational complexity analysis (Reviewer 6JF6, YxXK)

We will integrate this discussion following our method introduction: The time complexity of LECI is $O(md + nd^2)$ where $n$, $m$, and $d$ denote the number of nodes, edges, and feature dimension, respectively. To be more specific, the message-passing neural networks have time complexity $O(md+nd^2)$. Our environment exploitation regularizations have time complexity $O(1)$ without any extra cost. Therefore, the overall time complexity of our LECI is $O(md + nd^2 + 1) = O(md + nd^2)$.

### Explanations of the training dynamics in Figure 4. (Reviewer YxXK, psug)

During the initial phase of the training process depicted in Figure 4, independence constraints are minimally applied (or are not applied), ensuring a conducive environment for discriminator training. This approach is predicated on the notion that adversarial gradients yield significance only post the successful training of discriminators. The observed performance "drop" is attributed to the fact that the generalization optimization is yet to be activated, so it indicates the general subgraph discovery networks' performance.

This is a normal phenonmenon, and we can directly apply high independence constrains if we pretrain the discriminators.

## Additional experiments

- One large table for empirical comparisons on FIIF and PIIF synthetic datasets in the attached PDF file.
- Other experiments are included in one-to-one responses.

## Limitations of this work

The main limitations of this work are challenges that might not cover by this paper but **do not affect the main claims and conclusions of this paper**.

### The training difficulty in standard adversarial training.

We acknowledge the intrinsic complexities of adversarial training (characterized by bi-level optimization). However, two clarifications are in order: (1) refining the bi-level optimization process (e.g., the implementation of Wasserstein loss or consensus optimization) does not constitute the crux of our paper, and (2) despite these complexities, standard adversarial training has still rendered robust results, vindicating our claims. We've documented this limitation in section C.4 of our appendix.


### The accessibility and quality of environment labels.

Employing simple grouping strategies may inadvertently introduce noise into environment labels. Yet, our results unequivocally demonstrate that even such "noisy" environment labels surpass the efficacy of current sophisticated inductive biases. To obtain deeper insights into this comparison, please refer to section B.2 in our appendix. This observation, in turn, reinforces the resilience of our primary claims.

--------------------------------------

We're deeply appreciative of the insightful comments and open questions presented. We're eager to delve into these during our one-to-one rebuttal discussions. It's pertinent to note that while many of the questions are indeed thought-provoking and present fascinating challenges, the expansive nature of some may be beyond the scope of this singular piece of work. We humbly request that this be taken into consideration during the paper's reevaluation. Our sincere gratitude for the time invested and the invaluable feedback.

Sincerely,

Authors

---

### Decision · Program_Chairs · 2023-09-21

**Decision:**

Accept (poster)

**Comment:**

This paper addresses the problem of graph out-of-distribution generalization by incorporating label and environment causal independence simultaneously. I thank the authors and reviewers for the engaging and very comprehensive discussions.

The reviewers have highlighted that the paper is technically sound; that the illustration of graph OOD generalization from a causal perspective is clear and easy to understand; that theoretical results are strong and rigorous, showing that the derived optimal parameters can lead to learning causal subgraph only for invariant prediction;  and that the empirical evaluation is comprehensive, with the proposed approach outperforming existing methods by a large margin.

Some weaknesses were pointed out, for example, regarding computational complexity. I believe these have been addressed by the authors satisfactorily. The main critical point, however, is that of novelty, which was raised by one of the reviewers (psug). Novelty is largely a subjective criterion and none of the other reviewers either expressed a concern or praised the paper about it. Furthermore, Reviewer psug did not express additional _specific_ concerns wrt this issue after the authors’ rebuttal. I believe the authors have clarified how the paper relates to previous work (e.g., [1-2]) and how the paper contributes to the literature wrt other methods that have used environment-based approaches.

Overall, I believe this paper has enough contributions to be presented at NeurIPS and I trust the authors’ judgment to make the necessary changes, including the material presented at the rebuttal and in the appendix (esp. wrt to clarity on the novelty and relation to previous works).